# The GATOR complex regulates an essential response to meiotic double-stranded breaks in *Drosophila*

Youheng Wei[1,2†], Lucia Bettedi[1†], Chun-Yuan Ting[1], Kuikwon Kim[1], Yingbiao Zhang[1], Jiadong Cai[2], Mary A Lilly[1]*

[1]Cell Biology and Neurobiology Branch, National Institute of Child Health and Human Development, National Institutes of Health, Bethesda, United States; [2]College of Bioscience and Biotechnology, Yangzhou University, Yangzhou, China

**Abstract** The TORC1 regulator GATOR1/SEACIT controls meiotic entry and early meiotic events in yeast. However, how metabolic pathways influence meiotic progression in metazoans remains poorly understood. Here we examine the role of the TORC1 regulators GATOR1 and GATOR2 in the response to meiotic double-stranded breaks (DSB) during *Drosophila* oogenesis. We find that in mutants of the GATOR2 component *mio*, meiotic DSBs trigger the constitutive downregulation of TORC1 activity and a permanent arrest in oocyte growth. Conversely, in GATOR1 mutants, high TORC1 activity results in the delayed repair of meiotic DSBs and the hyperactivation of p53. Unexpectedly, we found that GATOR1 inhibits retrotransposon expression in the presence of meiotic DSBs in a pathway that functions in parallel to p53. Thus, our studies have revealed a link between oocyte metabolism, the repair of meiotic DSBs and retrotransposon expression.
DOI: https://doi.org/10.7554/eLife.42149.001

*For correspondence:
lillym@helix.nih.gov

†These authors contributed equally to this work

Competing interests: The authors declare that no competing interests exist.

## Introduction

We are interested in understanding how metabolism impacts meiotic progression during oogenesis. Target of Rapamycin Complex 1 (TORC1) is a multi-protein complex that functions as a master regulator of metabolism (*Loewith and Hall, 2011*; *Laplante and Sabatini, 2012a*; *Jewell and Guan, 2013*). In the presence of adequate nutrients and positive upstream growth signals, TORC1, which contains the serine/threonine kinase Target of Rapamycin, becomes active and functions to stimulate growth and inhibit catabolic metabolism through the phosphorylation of down-stream effector proteins. The Seh1 Associated Complex Inhibits TORC1 (SEACIT), originally identified in yeast, inhibits TORC1 activity in response to amino acid limitation (*Neklesa and Davis, 2009*; *Dokudovskaya et al., 2011*; *Wu and Tu, 2011*; *Bar-Peled et al., 2013*; *Panchaud et al., 2013*). SEACIT, known as the GAP Activity Towards Rags complex 1 (GATOR1) in metazoans, is comprised of three highly conserved proteins Npr2/Nprl2, Npr3/Nprl3 and Iml1/Depdc5 (*Bar-Peled et al., 2013*; *Panchaud et al., 2013*). In *Drosophila* and mammals, depleting any of the three GATOR1 components results in increased TORC1 activity and growth, as well as a reduced response to amino acid starvation (*Kowalczyk et al., 2012*; *Bar-Peled et al., 2013*; *Wei et al., 2014*; *Dutchak et al., 2015*; *Cai et al., 2016*; *Marsan et al., 2016*). Thus, the role of the SEACIT/GATOR1 complex in the regulation of TORC1 activity is highly conserved in eukaryotes.

The multi-protein GATOR2 complex, known as Seh1 Associated Complex Activates TORC1 (SEACAT) in yeast, inhibits the activity of GATOR1 and thus functions to activate TORC1 (*Bar-Peled et al., 2013*; *Wei et al., 2014*) (*Figure 1A*). In metazoans, the GATOR2 complex functions in multiple amino acid sensing pathways (*Bar-Peled et al., 2013*; *Panchaud et al., 2013*; *Chantranupong et al., 2014*; *Parmigiani et al., 2014*; *Kim et al., 2015*; *Cai et al., 2016*). In tissue

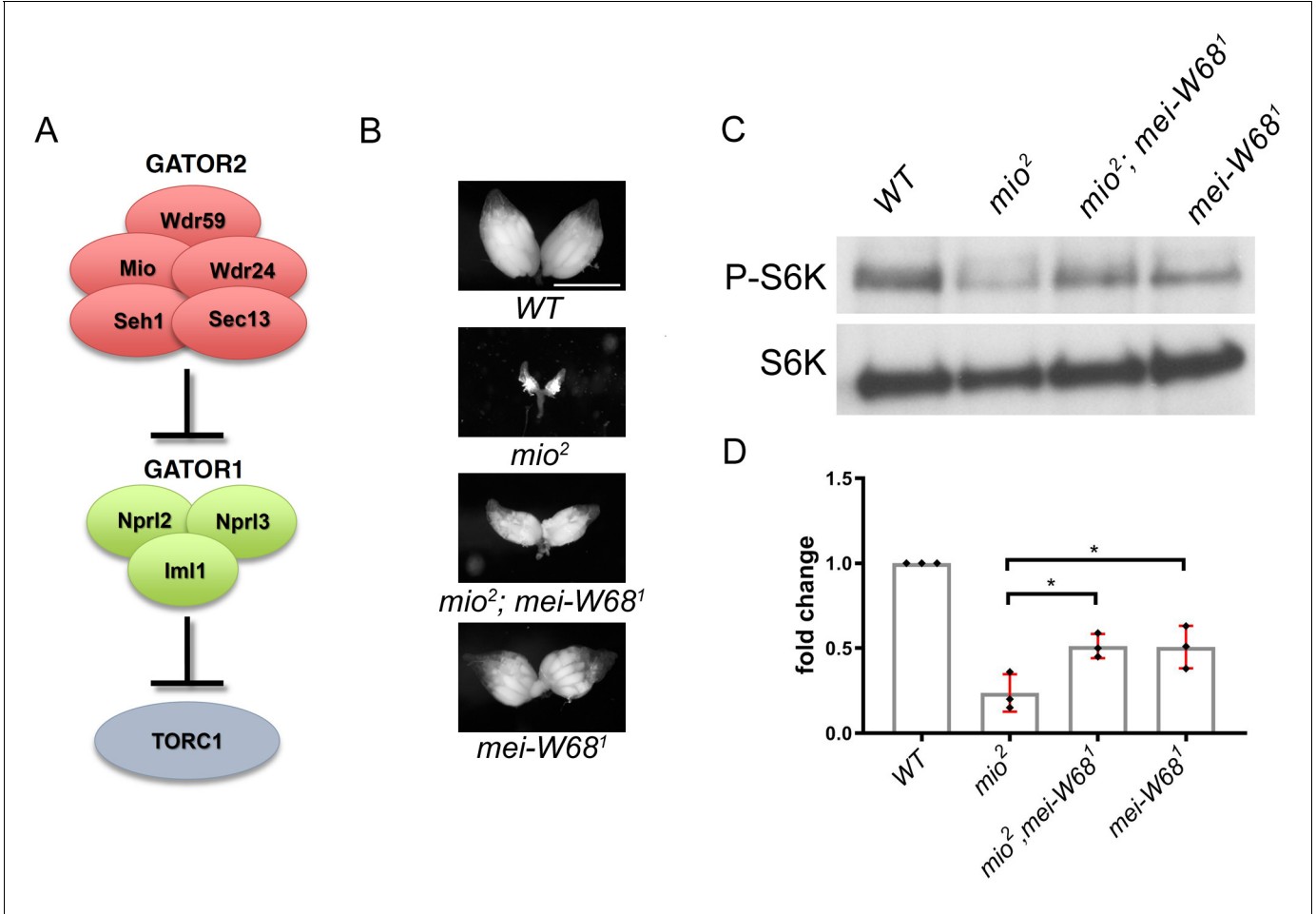

**Figure 1.** Mio prevents the constitutive downregulation of TORC1 activity in response to meiotic DSBs. (**A**) The GATOR2 complex opposes the activity of the TORC1 inhibitor GATOR1. (**B**) Representative ovaries from wild type (WT), $mio^2$, double-mutant $mio^2$, $mei-W68^1$ and $mei-W68^1$ females. Scale bar, 1000 μm. (**C**) Western blot of p-S6K and total-S6K levels of whole ovaries prepared from WT, $mio^2$ and $mio^2$, $mei-w68^1$ and $mei-W68^1$ mutant females. (**D**) Quantification of p-S6K levels relative to total S6K. Unpaired student T-test was used to calculate the statistical significance. Error bars represent the standard deviation (SD) for three independent experiments. *$p<0.05$.

DOI: https://doi.org/10.7554/eLife.42149.002

The following figure supplements are available for figure 1:

**Figure supplement 1.** Removing meiotic DSBs partially rescues the low egg production of *mio* mutants.
DOI: https://doi.org/10.7554/eLife.42149.003

**Figure supplement 2.** TORC1 activity is reduced in *spnA/Rad51* mutants.
DOI: https://doi.org/10.7554/eLife.42149.004

**Figure supplement 3.** Blocking the formation of meiotic DSBs fails to increase total TORC1 activity in wild type or *nprl3* mutant ovaries as measured by western blot.
DOI: https://doi.org/10.7554/eLife.42149.005

**Figure supplement 4.** Mutations in the checkpoint protein *loki* rescues the *seh1* ovarian phenotype.
DOI: https://doi.org/10.7554/eLife.42149.006

culture cells, depleting GATOR2 components results in the constitutive activation of GATOR1 and the permanent downregulation of TORC1 activity (*Bar-Peled et al., 2013*; *Wei and Lilly, 2014*). However, genetic studies of the role of individual GATOR2 components in *Drosophila*, indicate that the requirement for the GATOR2 complex is more nuanced when examined in the context of a multicellular animal (*Iida and Lilly, 2004*; *Wei et al., 2016*). For example, mutations in the GATOR2 component *mio*, result in a block to oocyte growth and differentiation, due to the constitutive downregulation of TORC1 activity in the female germline (*Iida and Lilly, 2004*; *Wei et al., 2016*).

However, *mio* is not required to maintain TORC1 activity in most somatic tissues of *Drosophila* (*Wei et al., 2016*). Why there is a tissue specific requirement for *mio* in the female germline of *Drosophila* is currently unknown.

In single celled eukaryotes, nutrient limitation often facilitates meiotic entry (*van Werven and Amon, 2011*). In the yeast *Saccharomyces cerevisiae*, the down-regulation of TORC1 by SEACIT/GATOR1 in response to amino acid stress promotes both meiotic entry and early meiotic progression (*Deutschbauer et al., 2002*; *Jordan et al., 2007*; *Neklesa and Davis, 2009*; *Spielewoy et al., 2010*). Surprisingly, as is observed in yeast, during *Drosophila* oogenesis the GATOR1 complex promotes meiotic entry (*Wei et al., 2014*). These data raise the intriguing possibility that in *Drosophila* the GATOR1 complex and low TORC1 activity may be critical to the regulation of additional events of the early meiotic cycle.

Here we report that the GATOR complex is critical to the response to meiotic DSB during *Drosophila* oogenesis. We find that restraining TORC1 activity via a pathway that involves both GATOR1 and the Tuberous sclerosis complex (TSC) promotes the timely repair of meiotic DSBs and prevents the hyperactivation of p53 in the female germline. Notably, the delayed repair of meiotic DSBs in GATOR1 mutants is due, at least in part, to the hyperactivation of the TORC1 target S6K. Conversely, our data indicate that the GATOR2 component Mio opposes the activity of GATOR1 in the female germline, thus preventing the constitutive downregulation of TORC1 activity and allowing for the growth and development of the oocyte in later stages of oogenesis. Thus, we have identified a regulatory loop required to modulate TORC1 activity in response to meiotic DSBs during *Drosophila* oogenesis. Finally, during the course of our studies, we observed that the GATOR1 complex prevents the derepression of retrotransposon expression in the presence of meiotic DSBs.

## Results

### Mio prevents the constitutive inhibition of TORC1 activity in response to meiotic DSBs

The GATOR2 complex inhibits the TORC1 inhibitor GATOR1 (*Figure 1A*). Ovaries from mutants of the GATOR2 component *mio,* have reduced TORC1 activity and are severely growth restricted (*Figure 1B–D*) (*Iida and Lilly, 2004*; *Wei et al., 2014*). In our previous studies, we demonstrated that the *mio* ovarian phenotypes result from the constitutive downregulation of TORC1 activity via a GATOR1 dependent pathway (*Wei et al., 2014*). Thus, removing GATOR1 activity in the *mio* mutant background, as is observed in *mio, nprl3* double mutants, results in increased TORC1 activity and rescues the *mio* ovarian phenotypes (*Wei et al., 2014*).

Surprisingly, *mio* mutants are also suppressed by blocking the formation of meiotic DSBs, with approximately 70% of *mio* ovaries achieving wild-type levels of growth when double mutant for genes required to generate meiotic DSBs (*Figure 1B*) (*Iida and Lilly, 2004*). One model to explain this observation is that meiotic DSBs promote the downregulation of TORC1 activity in the early meiotic cycle and that *mio* is required to oppose or attenuate this response. To test this idea, we examined if blocking the formation of meiotic DSBs in the *mio* mutant background resulted in increased TORC1 activity. Towards this end, we compared the phosphorylation status of S6 kinase, a downstream TORC1 target, in ovaries from *mio²* single mutant versus *mio, mei-W68* double mutant ovaries (*Figure 1C,D*). For these experiments, we used null alleles of both *mio (mio²)* and *mei-W68 (mei-W68¹)* (*McKim and Hayashi-Hagihara, 1998*; *Iida and Lilly, 2004*). *mei-W68 (SPO11* homolog) is a highly-conserved enzyme required for the generation of meiotic DSBs (*McKim and Hayashi-Hagihara, 1998*; *Sekelsky et al., 1999*; *Liu et al., 2002*). We found that relative to ovaries from *mio* single mutants, *mio, mei-W68* double mutants have increased levels of TORC1 activity as measured by the phosphorylation of S6K (*Figure 1C,D*). Notably, *mio²*, *mei-W68¹* mutants have TORC1 activity levels similar to that observed in *mei-W68¹* single mutants (*Figure 1C,D*). Why ovaries from *mei-W68¹* single mutants have decreased levels of TORC1 activity relative to wild type ovaries is unclear. In addition to increasing TORC1 activity, blocking the formation of meiotic DSBs partially rescues *mio* mutant fertility, with *mio²*, *mei-W68¹* double mutants laying approximately ten times more eggs than *mio²* single mutants (*Figure 1—figure supplement 1*). Finally, mutants in the *spnA* homolog Rad51, which fail to repair meiotic DSBs, also have decreased TORC1 activity relative to wild-type ovaries (*Figure 1—figure supplement 2*). From these data, we conclude that the constitutive

downregulation of TORC1 activity in *mio* mutants is potentiated, at least in part, by the presence of meiotic DSBs.

To refine when meiotic DSBs impact TORC1 activity during meiosis, we stained ovaries with antibodies against the phosphorylated form of the TORC1 target 4E-BP (*Teleman et al., 2005*). TORC1-mediated phosphorylation of 4E-BP, known as Thor in Drosophila, initiates cap-dependent translation by eIF4E (*Gingras et al., 1998*) (*Teleman et al., 2005*). In *Drosophila*, oogenesis begins in region 1 of the germarium when a germline stem cell divides to produce a cystoblast that undergoes four mitotic divisions, with incomplete cytokinesis, to produce a 16 cell interconnected germline cyst (*Figure 2A*) (*de Cuevas et al., 1997*). In late region 2a of the germarium, Spo11/Mei-W68 generates the meiotic DSBs that initiate meiotic recombination (*McKim and Hayashi-Hagihara, 1998*; *Hunter, 2015*). As meiosis proceeds, meiotic DSBs are repaired such that by late region 2b, only a small fraction of oocytes retain unrepaired DSBs (*Mehrotra and McKim, 2006*; *Narbonne-Reveau and Lilly, 2009*). We found that the levels of p4E-BP are low in the vast-majority of ovarian cysts undergoing mitotic divisions in region 1 (*Figure 2A',B*). However, a small number (0.5 per germarium, n = 57) of region one ovarian cysts have dramatically higher levels of p4E-BP staining (*Figure 2A"*, arrowhead). These observations are consistent with the previously reported oscillation of TORC1 activity during the mitotic cell cycle in larval imaginal discs (*Kim et al., 2017*; *Romero-Pozuelo et al., 2017*). As ovarian cysts enter meiosis I in early region 2a of the germarium, the levels of p4E-BP are low (*Figure 2A',B*). As ovarian cysts are encapsulated by follicle cells in germarial

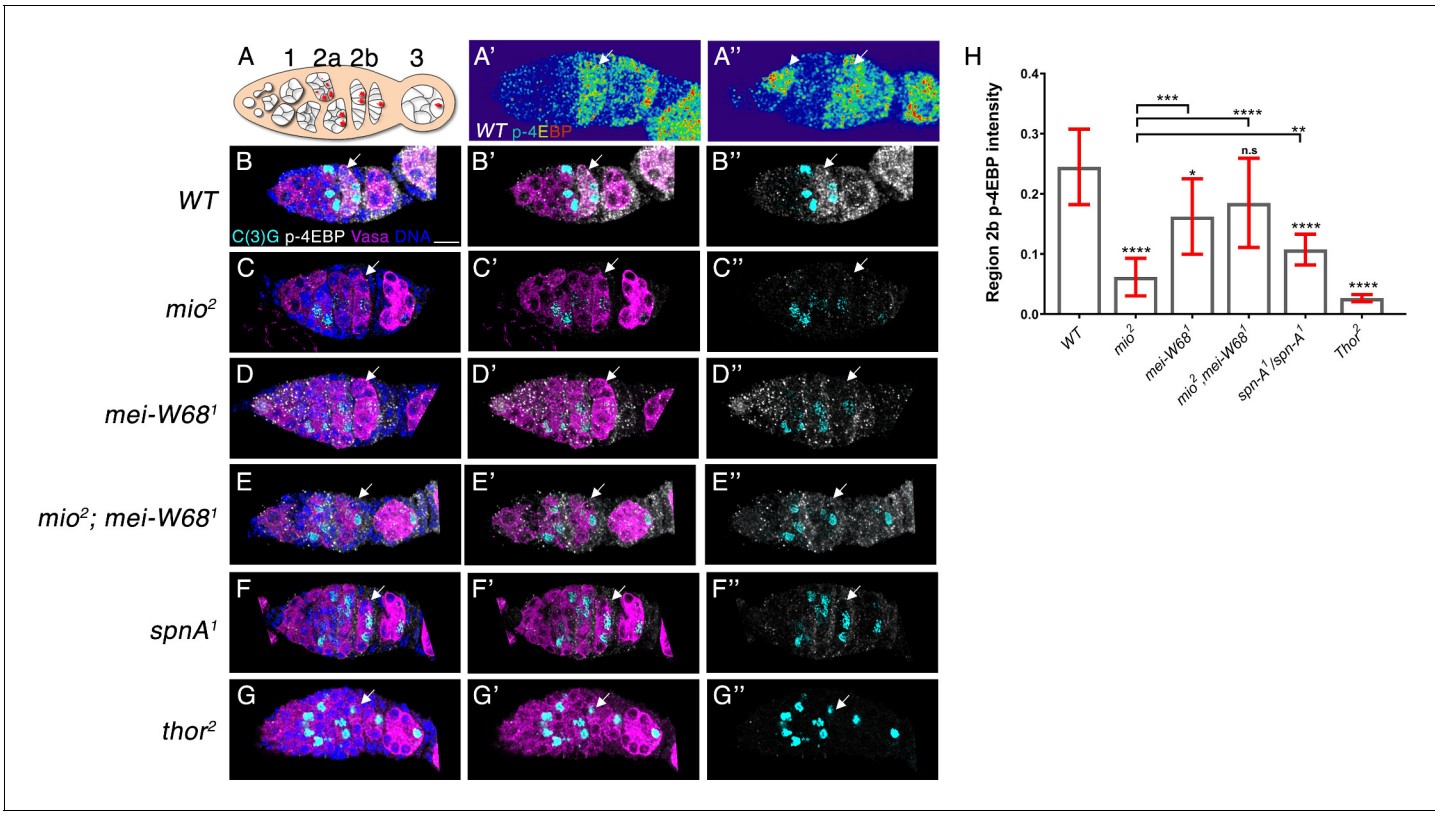

**Figure 2.** Suppressing the production of meiotic DSBs increases p4E-BP staining in the female germline of *mio* mutants. (**A**) Schematic representation of the *Drosophila* germarium. (**A'A'**) pseudo-color representation of p-4E-BP staining, arrowhead denotes a region one ovarian cyst with high p-4E-BP levels. Ovaries from (**B**) wild type (**C**) *mio²*, (**D**) *mei-W68¹*, (**E**) *mio²; mei-W68¹*, (**F**) *spnA¹*, (**G**) *thor²* females stained for C(3)G (cyan) to mark the synaptonemal complex, p-4E-BP (white), Vasa (magenta) to highlight the germline cytoplasm and DNA (Blue). (**B**) In wild-type ovarian cysts, p-4E-BP staining begins to increase in region 2b (arrow). (**C**) In *mio* mutant ovarian cysts, p-4E-BP levels remain low in region 2b and region 3. (**D,E**) *mio*, *mei-W68* double mutants have p-4E-BP expression levels similar to *mei-W68* single mutants. (**F**) *spnA* mutants, which fail to repair meiotic DNA breaks, have low levels of p-4E-BP staining (**G**) *thor²*/4E-BP null mutants serve as a negative control. (**H**) p-4E-BP intensity measurement of region 2b data (**B**)-(**G**). Scale bar: 7 μm. Unpaired T-student test was used to calculate statistical significance. *p<0.05, **p<0.01, ***p<0.001, ****p<0.0001.
DOI: https://doi.org/10.7554/eLife.42149.007

region 2b, the levels of p4E-BP begin to rise and remain above those observed in region one and early region 2a (*Figure 2B*, arrow). In contrast to what is observed in wild type, in *mio* mutant ovarian cysts, p4E-BP levels remain low in germline cells in region 2b and beyond (*Figure 2C*, arrow). Consistent with the western blot analysis, *mio, mei-w68* double mutant ovarian cysts have an approximately three-fold increase in p4E-BP staining in region 2b of the germarium relative to *mio* mutants (*Figure 2C,E and H*) Notably, the increase in TORC1 activity in the *mio, mei-w68* double mutants is restricted to the germline, consistent with blocking meiotic DSBs having cell autonomous effects on TORC1 activity in the germline. Additionally, consistent with our western blot analysis in *Figure 1—figure supplement 2*, ovarian cysts from *spnA/Rad51* mutants, which do not repair meiotic DSBs, have reduced levels of p4E-BP staining (*Figure 2F*). Taken together these data strongly suggest that *mio* is required to oppose the downregulation of TORC1 activity triggered by the presence of meiotic DSBs.

To examine if meiotic DSBs activate the downregulation of TORC1 activity via GATOR1, or activate a parallel TORC1 inhibitory pathway, we performed epistasis analysis with a null mutant of the GATOR1 component *nprl3* (*nprl3[1]*) and a mutant of *mei-P22*, a gene required for the generation of meiotic DSBs (*McKim et al., 1998*; *Liu et al., 2002*; *Iida and Lilly, 2004*). The *mei-P22[P22]* mutation, which rescues the *mio* ovarian growth deficit, blocks the formation of meiotic DSBs resulting in meiotic exchange rates of zero or near zero (*McKim et al., 1998*; *Liu et al., 2002*; *Iida and Lilly, 2004*). We found that *mei-P22[P22]* single mutants had pS6K levels that were not significantly different than ovaries from wild-type females when measured by western blot (*Figure 1—figure supplement 3*). This result is not surprising when one considers the anatomy of the *Drosophila* ovary. In wild-type ovaries, meiotic DSBs are present in only a small number of 16 cell cysts in the germarium. Moreover, meiotic DSBs are repaired prior to the rapid growth of the egg chamber. Thus, in wild type females, ovarian cysts that contain meiotic DSBs represent an exceedingly small percentage of the tissue in the ovary. Therefore, it is unlikely that increasing TORC1 activity in only a small number of germarial ovarian cysts would result in an increase in TORC1 activity in the ovary that could be observed by western blot. In contrast, note that ovaries from *spnA* mutants, which retain DSBs throughout oogenesis, have low TORC1 activity relative to wild type ovaries (*Figure 1—figure supplement 2*). Consistent with *mei-P22* mutant ovaries not having increased TORC1 activity, we found that *mei-P22, nprl3* double-mutant ovaries do not have pS6K levels above those of *nprl3* single mutants. Thus, using epistasis analysis we were unable to definitely determine if meiotic DSBS trigger the downregulation of TORC1 activity via the GATOR1/TSC pathway.

We were interested in defining the upstream pathway that connects meiotic DSBs to the TORC1 regulatory machinery. *Ataxia telangiectasia* (ATM), known as *telomere fusion* (*tefu*) in *Drosophila*, regulates both the generation and the repair of meiotic DSBs (*Joyce et al., 2011*). *atm/tefu* mutants produce supernumerary meiotic DSBs. Therefore, we examined a downstream target and effector of *atm/tefu*, *chk2*, which has not been implicated in the generation of meiotic DSBs. *chk2*, known as *loki* in *Drosophila*, is a critical component of the DNA damage response pathway and has multiple targets involved in DNA repair, cell cycle progression and apoptosis (*Zannini et al., 2014*). We determined that removing *chk2/loki* activity in the *mio* mutant background, by generating double-mutants of *loki* and *mio* null alleles, partially rescued the *mio* mutant phenotype (*Figure 1—figure supplement 4*). Specifically, we found that *mio[2], loki[6]* double mutants are approximately twice the size of *mio[2]* single mutants. In contrast, we previously demonstrated that the downregulation of TORC1 activity observed in the *mio* mutant is not triggered by the upstream activity of Ataxia Telangiectasia–Related (ATR), known as *mei-41* in *Drosophila* (*Iida and Lilly, 2004*; *Joyce et al., 2011*). These data suggest that DSBs communicate to the TORC1 machinery at least in part through the checkpoint protein Chk2/Loki.

## GATOR1 promotes the repair of meiotic DSB

In previous work we found that the GATOR1 complex downregulates TORC1 activity to facilitate meiotic entry in *Drosophila* ovarian cysts (*Deutschbauer et al., 2002*; *Jordan et al., 2007*; *Neklesa and Davis, 2009*; *Spielewoy et al., 2010*; *Wei et al., 2014*). However, the delay in meiotic entry observed in *Drosophila* GATOR1 mutants is not fully penetrant and therefore unlikely to be the sole cause of the infertility observed in GATOR1 mutant females (*Figure 3—figure supplement 1A,B*) (*Wei et al., 2014*). Considering our findings that meiotic DSBs serve to promote and/or reinforce low TORC1 activity after the mitotic/meiotic switch, we hypothesized that as is observed in

yeast, the downregulation of TORC1 activity may be critical to the regulation of additional early meiotic events, including the repair of meiotic DSBs in *Drosophila*.

To test this hypothesis, we examined the behavior of meiotic DSBs in null alleles of *nprl2*, *nprl3* and *iml1* (*Cai et al., 2016*; *Wei et al., 2016*). During *Drosophila* oogenesis, the kinetics of DSB formation and repair can be followed using an antibody against the phosphorylated form of His2Av known as γ-His2Av (*Madigan et al., 2002*; *Mehrotra and McKim, 2006*). *Drosophila* ovarian cysts generate meiotic DSBs after the initiation of synaptonemal complex (SC) formation in region 2a of the germarium (*Carpenter, 1975*; *Jang et al., 2003*; *Mehrotra and McKim, 2006*). γ-H2Av nuclear foci are first observed in the two pro-oocytes, which are in early pachytene (*Jang et al., 2003*; *Mehrotra and McKim, 2006*). A small number of DSBs are also observed in the pro-nurse cells (*Mehrotra and McKim, 2006*; *Narbonne-Reveau and Lilly, 2009*). As meiosis proceeds and the DSBs are repaired, γ-H2Av-positive foci decrease in number and mostly disappear by late region 2b (*Jang et al., 2003*; *Mehrotra and McKim, 2006*). γ-H2Av signals are rarely detected in region three oocytes (*Figure 3A*, arrow). Analysis of mutants that fail to repair DSBs, and thus capture the total number of meiotic breaks, indicate that wild-type oocytes generate approximately 20–25 Spo11/Mei-W68 dependent breaks during oogenesis (*Mehrotra and McKim, 2006*; *Joyce et al., 2011*).

To determine if GATOR1 regulates the behavior of meiotic DSBs in *Drosophila*, we compared the pattern of γ-H2Av foci in wild-type versus GATOR1 mutant ovaries using antibodies against γ-H2Av and the SC component C(3)G, to highlight DSBs and oocytes respectively (*Iida and Lilly, 2004*; *Mehrotra and McKim, 2006*). We determined that while the majority of wild-type oocytes had repaired all of their DSBs and thus had no γ-H2Av foci by region 3 of the germarium, in GATOR1 mutants between 50–80% of region three oocytes are γ-H2Av positive (*Figure 3A–E,H*, arrow) (*Mehrotra and McKim, 2006*). Moreover, GATOR1 mutant oocytes had a significant increase in the steady state number of γ-H2Av foci per oocyte nucleus in region 2a of the germarium relative to wild-type oocytes (*Figure 3J*). From these data, we conclude that the GATOR1 complex influences the behavior of meiotic DSB during early oogenesis.

Next, we examined if the altered γ-H2Av pattern observed in GATOR1 mutants was dependent on the meiotic DSB machinery. Alternatively, the extra DSB may be induced during the premeiotic S phase, as is observed in mutants of the CycE/Cdk2 inhibitor *dacapo* (*Hong et al., 2003*). To address this question, we analyzed *nprl3*, *mei-P22* double-mutants. As discussed above, *mei-P22* is required for the formation of meiotic DSBs in *Drosophila* (*Liu et al., 2002*). We determined that double-mutant *nprl3*, *mei-P22* oocytes had no γ-H2Av foci (*Figure 3F,I and J*). These data indicate that the increase in the steady state number of γ-H2Av foci, as well as the retention of these foci into region 3 of the germarium, are dependent on the meiotic DSB machinery.

During the development of wild-type *Drosophila* oocytes, the generation and repair of meiotic DSBs is asynchronous. Thus, the number of γ-H2Av foci observed in an oocyte at any single time point, is less than the total number of DSBs generated during the lifetime of the oocyte.

(*Mehrotra and McKim, 2006*; *Joyce et al., 2011*). We noticed that the number of γ-H2Av foci observed in GATOR1 mutant oocytes is never more than the 20–25 foci observed in mutants in the DSB repair pathway (*Figure 3J*) (*Mehrotra and McKim, 2006*). This observation suggested that the increase in the steady state number of DSBs observed in GATOR1 mutants may result from the delayed repair of meiotic DSBs, rather than the production of supernumerary Mei-W68/Spo11 induced DSBs. To test this hypothesis, we generated *nprl2*, *spnA* double mutants. Importantly, *spnA* mutants, including the *spnA[1]/spnA[093]* transheterozygotes used here, fail to repair meiotic DSBs (*Figure 3G*) (*Staeva-Vieira et al., 2003*; *Joyce and McKim, 2011*). If *nprl2* mutants make extra meiotic breaks, then the number of foci in the *nprl2*, *spnA* double mutants should be higher than either single mutant. However, we found that the *nprl2*, *spnA* double-mutants contained approximately the same number of γ-H2Av foci as *spnA* single mutants (*Figure 3K*). Thus, mutations in *nprl2* do not result in the production of extra meiotic breaks. Taken together, our data strongly suggest that the GATOR1 complex influences the repair, rather than the production, of meiotic DSBs.

In *Drosophila*, the failure to repair meiotic DSBs activates an ATR-dependent checkpoint that disrupts dorsal ventral (DV) patterning in the egg (*Ghabrial et al., 1998*; *Ghabrial and Schüpbach, 1999*). We find that approximately 98% of eggs from *nprl3[1]* mothers exhibit no DV patterning defects (n = 308). Moreover, the approximately 2% of eggs from *nprl3* mothers that have a possible DV patterning defect, are smaller and have a collapsed egg shell. Thus, the DV patterning defects observed in eggs from *nprl3* mutant mothers may reflect a general problem in egg chamber growth

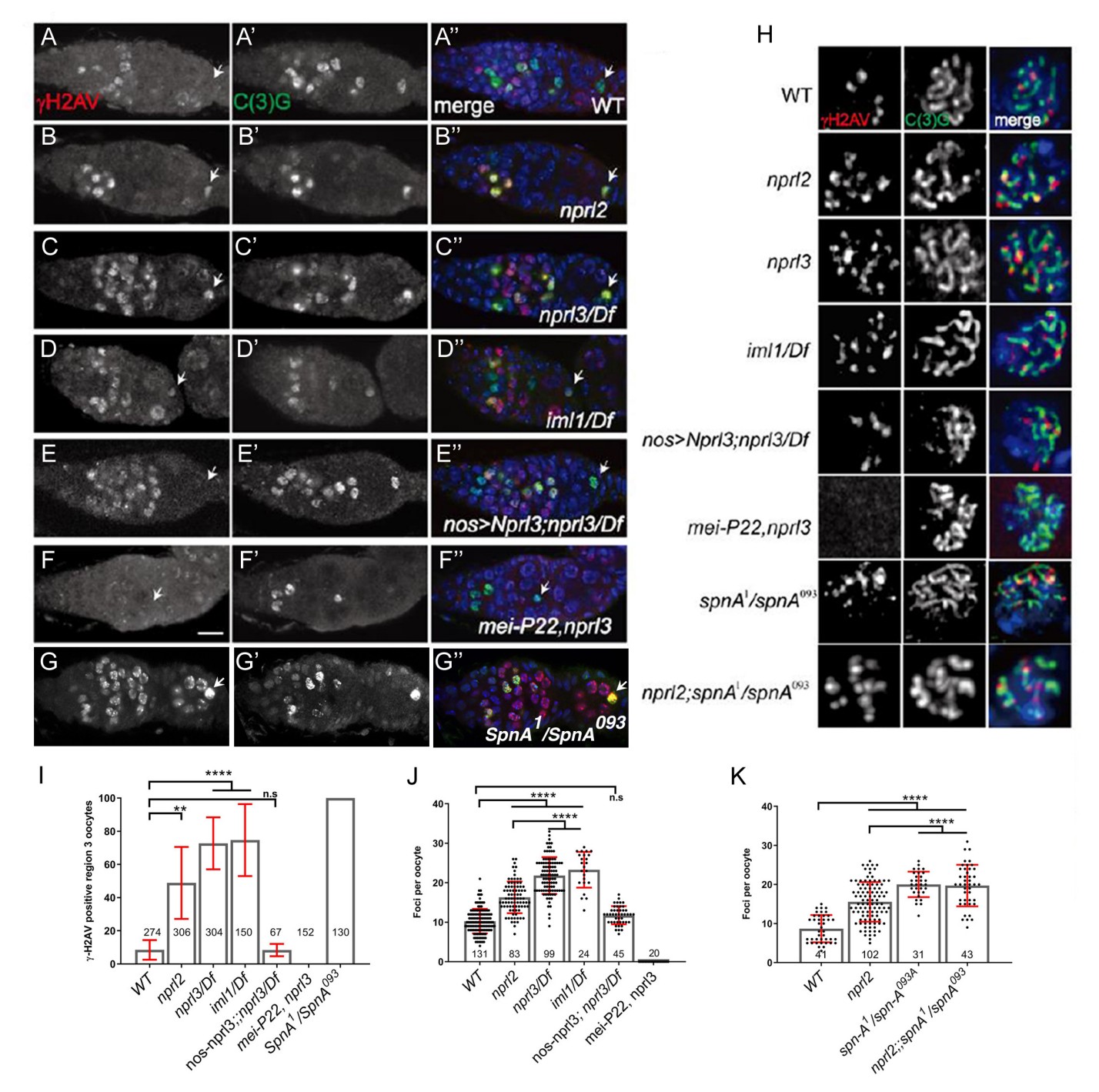

**Figure 3.** GATOR1 influences the steady state number and persistence of DSBs in early oocytes. Ovaries from (**A**) wild type, (**B**) *nprl2[1]*, (**C**) *nprl3[1]/Df*, (**D**) *iml1[1]/Df*, (**E**) *nanos-GAL4; UAS-Nprl3; nprl3[1]/Df*, (**F**) *mei-P22[P22]*, *nprl3[1]*, and (**G**) *spnA[1]/spnA[093]* females were stained for C(3)G (green, **A'–G'**) and γ-H2Av (red, **A–G**). C(3)G marks the synaptonemal complex (SC) and is used to mark oocytes and follow meiotic progression. γ-H2Av marks DSBs. Scale bars, 10 μm. In wild type oocytes, meiotic DSBs are induced in region 2a and repaired by region 3 (arrow). In GATOR1 mutants, DSBs persist in region three oocytes. In *nanos-GAL4; UAS-Nprl3; nprl3[1]/Df* oocytes, DSBs are repaired by region 3. *mei-P22[P22]*, *nprl3[1]* mutants have no DSBs. (**H**) Ovaries from wild type, *nprl2[1]*, *nprl3[1]/Df*, *iml1[1]/Df*, *spn-A[1]/spn-A[093A]*, *nprl2[1]; spnA[1]/spnA[093]*,*mei-P22[p22]*,*nprl3[1]* and *nanos-GAL4; UAS-Nprl3; nprl3[1]/Df* flies were stained for C(3)G (green) and γ-H2Av (red). Representative immunofluorescent microphotographs of the γ-H2Av foci in region 2a oocyte are shown. (**I**) Percentage of region three oocytes with γ-H2Av foci. (**J and K**) Quantification of γ-H2Av foci in region 2a oocytes. Unpaired T-student test was used to calculate the statistical significance. Error bars represent SD from at least three independent experiments.**p<0.01, ****p<0.0001, ns: no significance. Note that the three GATOR1 mutants, *nprl3[1]*, *nprl2[1]* and *iml1[1]* are null alleles (*Cai et al., 2016*; *Wei et al., 2016*).

*Figure 3 continued on next page*

*Figure 3 continued*

DOI: https://doi.org/10.7554/eLife.42149.008

The following figure supplement is available for figure 3:

**Figure supplement 1.** *nprl2* and *nprl3* mutant females are semi-sterile.

DOI: https://doi.org/10.7554/eLife.42149.009

and development. In contrast, 77% of eggs from mutant *spnA[1]* mothers exhibit DV patterning defects (n = 271) (*Staeva-Vieira et al., 2003*). Why we do not observe DV patterning defects in GATOR1 mutants is unclear but may indicate that high TORC1 activity delays but does not block all aspects of meiotic DNA repair. Alternatively, high TORC1 activity may override the translational repression of the patterning gene *gurken* that drives the pattern defects observed in DNA repair mutants such as *spnA* (*Abdu et al., 2002*).

## Co-depleting S6K rescues the increase in the steady state number of meiotic DSBs in iml1 germline depletions

TORC1 stimulates protein synthesis and cell growth through the phosphorylation of downstream effector proteins that promote anabolism and inhibit catabolism (*Hay and Sonenberg, 2004*; *Wullschleger et al., 2006*). We wanted to identify the pathways downstream of TORC1 that impact the repair of meiotic DSBs (*Wei et al., 2016*). Towards this end we examined the role of three well known downstream targets of TORC1 phosphorylation, S6K, Atg1 and 4E-BP in the regulation of meiotic DSBs (*Laplante and Sabatini, 2012b*). We determined that the depletion of *iml1* by RNAi, recapitulates the *iml1* mutant phenotype resulting in an increase in the steady state number of meiotic DSBs in the germarium (*Figure 4*). TORC1 phosphorylation activates S6K, a serine–threonine

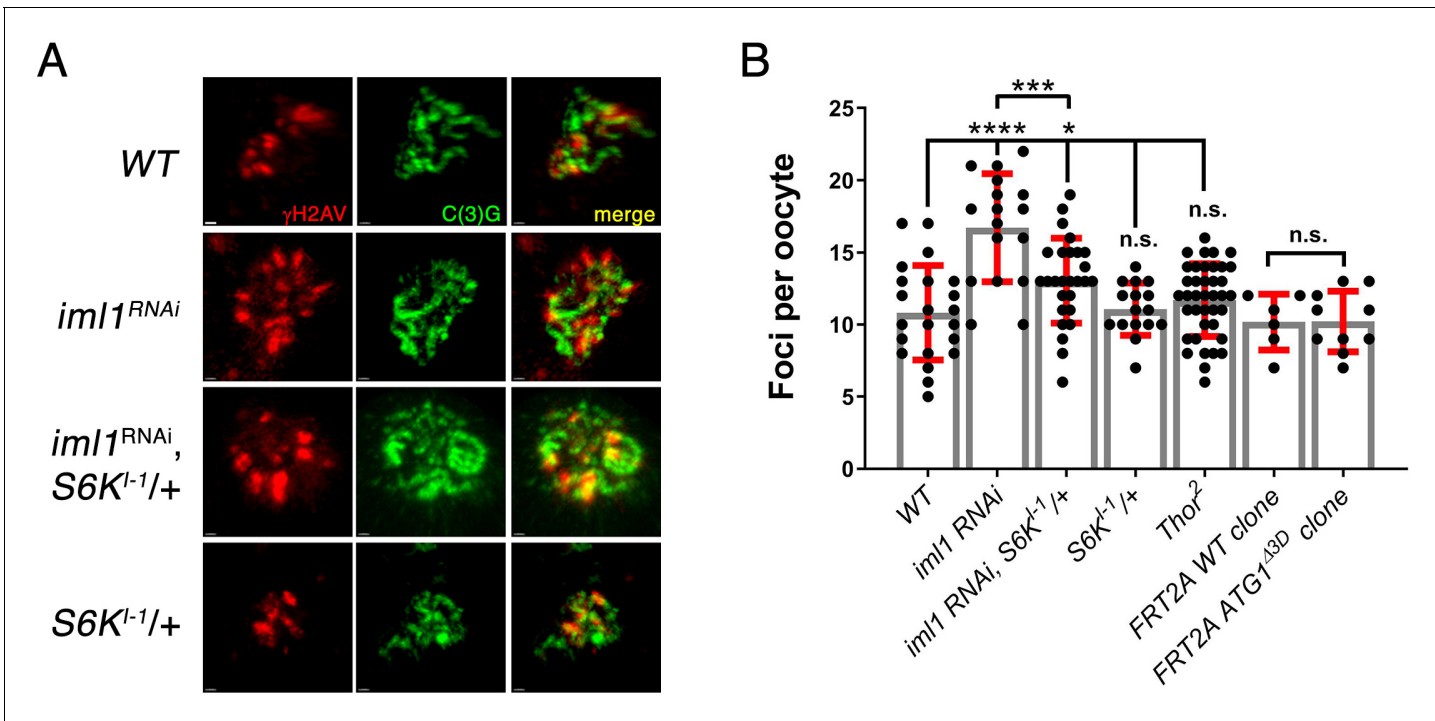

**Figure 4.** Reducing the dose of S6K rescues the meiotic DSB phenotype in *iml1* knockdowns. (A) γ-H2Av foci in region 2a oocytes in the indicated genotypes. (B) Quantification of γ-H2Av foci in region 2a oocytes in the indicated genotypes. Note that meiotic DSBs were increased in *nanos-GAL4; iml1*[RNAi] females. Moreover, removing a single copy of S6K in the *nanos-GAL4; iml1*[RNAi] background reduced the steady state number of meiotic DSBs in *nanos-GAL4; iml1*[RNAi]; *S6K*[l-1]/+ females. Unpaired T-student test was used to calculate statistical significance. *p<0.05, ****p<0.0001, n.s.: no significance. (B).

DOI: https://doi.org/10.7554/eLife.42149.010

kinase that promotes translation and growth (*Tavares et al., 2015*). In mammals, S6K links growth control to the DNA damage response (*Xie et al., 2018*). To determine if the hyperactivation of S6K contributes to the defects in the repair of meiotic DSBs observed in GATOR1 mutants, we reduced the dose of the S6K gene by half in the *iml1* germline depletions. Notably, reducing the dose of S6K lowered the steady state number of meiotic DSBs in *iml1* RNAi depletions to levels observed in controls (*Figure 4A,B*). These data indicate that the delay in the repair of meiotic DSBs observed in GATOR1 mutant ovaries is at least in part the result of the hyperactivation of S6K.

High TORC1 inhibits the activation of autophagy via the inhibitory phosphorylation of Atg1 and Atg13 (*Galluzzi et al., 2014*). In our previous work, we determined that GATOR1 mutants fail to undergo autophagy in response to starvation due to inappropriately high TORC1 activity (*Wei et al., 2016*). Recent evidence suggests that blocking autophagy inhibits DSB repair through homologous recombination (*Hewitt and Korolchuk, 2017*). Therefore, to examine if a block to autophagy is responsible for the delay in the repair of meiotic DSBs we generated germline clones of a null allele of *atg1*, which is required for the activation of autophagy in *Drosophila* (*Scott et al., 2004*). We found that late region 2a oocytes in *atg1*[Δ3D] germline clones had approximately the same number of γ-H2Av foci as similarly staged wild-type oocytes (*Figure 4B*). From these data, we conclude that the increased number of meiotic DSBs observed in GATOR1 mutants is unlikely to be the result of a block to autophagy. Similarly, we found that null mutants of the translational inhibitor 4E-BP/*thor*, which is inhibited by TORC1 activity do not have an increased steady state number of meiotic DSBs. These data argue that the increase in the steady state number of meiotic DSBs in GATOR1 mutants is not the result of increased translation due to a block to 4E-BP translational inhibition.

GATOR1 mutants hyperactivate p53 in response to meiotic DSBs p53, a transcription factor that mediates a conserved response to genotoxic stress, regulates early meiotic events in multiple organisms (*Stürzbecher et al., 1996*; *Lee et al., 1997*; *Linke et al., 2003*; *Mateo et al., 2016*). During *Drosophila* oogenesis, the generation of meiotic DSBs results in the brief activation of p53 and the expression of downstream targets (*Figure 5A*) (*Lu et al., 2010*). To determine if GATOR1 mutants experience increased genotoxic stress during oogenesis, we used a reporter construct to assay p53 activity. The p53-GFPnls reporter construct contains the Green Fluorescent Protein (GFP) under the control of an enhancer from the p53 transcriptional target *reaper* (*Lu et al., 2010*). In wild-type ovaries, a faint signal from the p53-GFPnls reporter is first observed in region 2a of the germarium, concurrent with the generation of meiotic DSBs (*Figure 5A*, arrow) (*Lu et al., 2010*). As ovarian cysts continue to develop, the p53-GFPnls signal rapidly dissipates as meiotic DSBs are repaired (*Figure 5A*, arrowhead) (*Lu et al., 2010*). By region 3 of the germarium less than 5% of p53-GFPnls ovarian cysts contain detectable levels of GFP (*Figure 5A*). In contrast, the germaria of all three GATOR1 mutants exhibited a dramatic increase in both the strength and the duration of p53-GFPnls expression in the germarium, with strong GFP signal observed in nearly 80% of region three ovarian cysts (*Figure 5A–D,F,G*, arrowheads). Homozygous germline clones of the *iml1*[1] and *nprl3*[1] null alleles, hyperactivate p53 confirming that GATOR1 activity is required cell autonomously in the female germline (*Figure 5—figure supplement 1*).

We predicted that the persistent hyperactivation of p53 in the female germ line of GATOR1 mutants is due to the delayed repair of meiotic DSBs. To test this model, we examined p53 activation in the *mei-P22, nprl3* double mutant. Strikingly, inhibiting meiotic DSBs strongly suppressed the expression of p53-GFPnls in the *nprl3* mutant background (*Figure 5E*). From these observations, we infer that in GATOR1 mutant ovaries, the hyperactivation of p53 is downstream of Spo11/Mei-W68-induced DBS. Moreover, we conclude that the GATOR1 complex is required to oppose genotoxic stress triggered by meiotic DSBs and/or other downstream events of meiotic recombination.

Recent evidence indicates that p53 is activated in germline stem cells of *Drosophila* after exposure to cellular stresses, including deregulated growth and ionizing radiation (*Wylie et al., 2016*; *Ma et al., 2016*). The GATOR1 complex inhibits TORC1 activity and is required to restrain growth in *Drosophila* (*Wei et al., 2014*; *Cai et al., 2016*). As was reported with other mutants that deregulate growth (*Wylie et al., 2016*), we found that the p53-GFPnls reporter construct is robustly activated in the germline stem cells and their near descendants in GATOR1 mutant females (*Figure 5B–D,H*). Consistent with the restriction of Mei-W68/Spo11 activity to meiotic cysts, the *mei-P22, nprl3* double mutants retain p53-GFPnls expression in stem cells even though the meiotic activation of the p53-GFPnls reporter is lost in regions 2a and 2b of the germarium in the *mei-P22, nprl3* double mutant

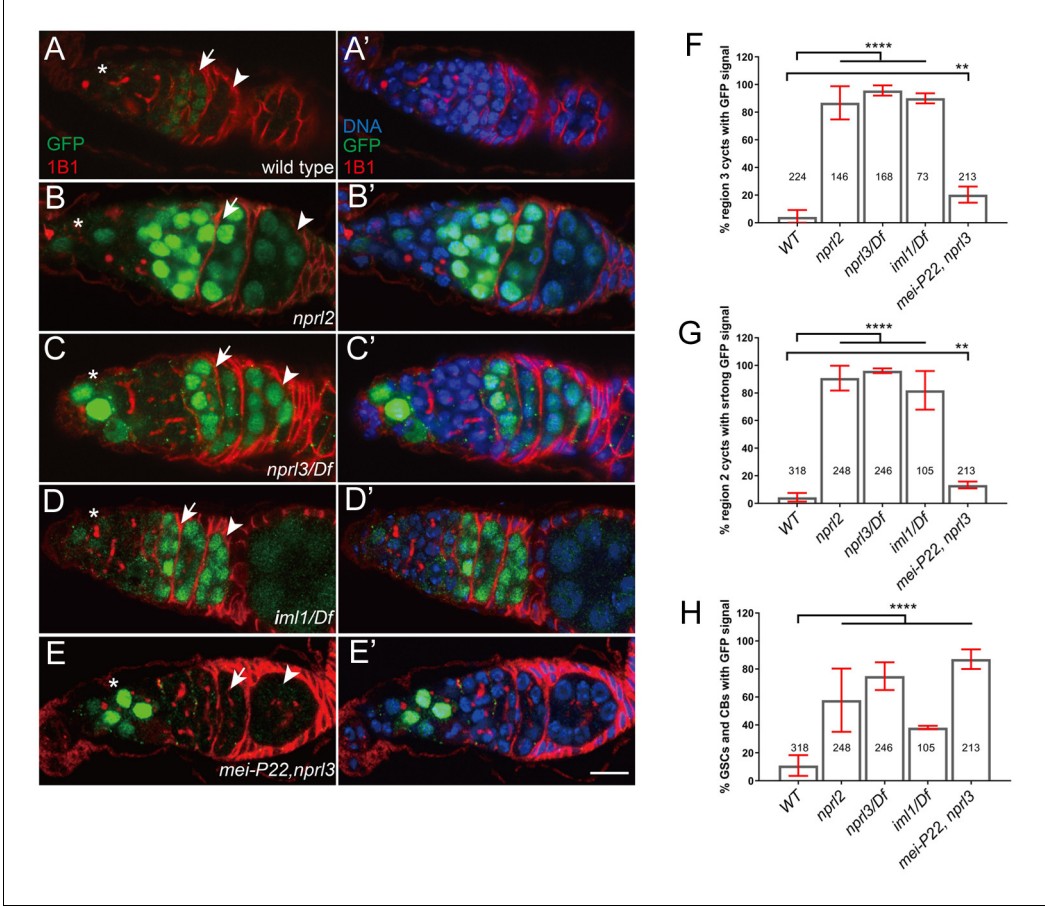

**Figure 5.** GATOR1 prevents p53 hyperactivation in Drosophila early ovarian cysts Ovaries from (**A**) *p53R-GFP*, (**B**) *nprl2[1]; p53R-GFP*, (**C**) *p53R-GFP;nprl3[1]/Df*, (**D**) *p53R-GFP; iml1[1]/Df* and (**E**) *p53R-GFP; mei-P22[p22], nprl3[1]* were stained for GFP (green) and 1B1 (red). Germarial regions are defined by 1B1 staining. In wild-type ovaries the p53-GFP reporter is briefly activated in region 2 (indicated by arrow). Note the low level of GFP staining. In contrast, in GATOR1 mutants, p53R-GFP is robustly activated with strong GFP signal often persisting into germarial region three and beyond. Additionally, in GATOR1 mutant germaria, p53R-GFP is frequently activated in germline stem cell (GSC) and daughter cystoblasts (CB). In *mei-P22[p22], nprl3[1]* double mutant germaria, the hyperactivation of p53R-GFP is rescued in region 2a ovarian cysts. However, p53-GFP activation in GSC and CB is retained in the double mutants (asterisk) indicating that in these cells the activation of p53 is not contingent on the presence of meiotic DSBs. Scale bars, 10 μm. (**F**) Percentage of germaria with sustained p53R-GFP signal in region 3. (**G**) Percentage of germaria with high p53R-GFP signal in region 2. (**H**) Percentage of germaria with p53R-GFP expression in GSC and CB. Unpaired student T-test was used to calculate the statistical significance Error bars represent SD from at least three independent experiments. **p<0.01, ****p<0.0001.

DOI: https://doi.org/10.7554/eLife.42149.011

The following figure supplement is available for figure 5:

**Figure supplement 1.** The GATOR1 complex acts cell autonomously in the female germ line.

DOI: https://doi.org/10.7554/eLife.42149.012

(*Figure 5E–G*). Thus, an independent cellular stress, likely related to deregulated metabolism or growth, activates p53 in the germline stem cells and cystoblasts of GATOR1 mutant females.

## Tsc1 germline depletions phenocopy GATOR1 mutants

The most parsimonious interpretation of our data is that in GATOR1 mutant ovaries, high TORC1 activity opposes the timely repair of meiotic DSBs and increases genotoxic stress. To test this model, we depleted the Tuberous sclerosis complex (TSC) component Tsc1 from the female germline. TSC is a potent inhibitor of TORC1 that directly inhibits the small GTPase Rheb, a critical activator of

TORC1 (*Inoki et al., 2003*; *Zhang et al., 2003*). We determined that depleting *Tsc1* in the female germline using RNAi, resulted in the robust expression of p53-GFPnls during the early meiotic cycle (*Figure 6A–D*). Moreover, as was observed in GATOR1 mutants, *Tsc1^{RNAi}* oocytes had an increase in the steady state number of γ-H2Av foci as well as an increase in the percentage of oocytes that retained γ-H2Av positive into region 3 of the germarium (*Figure 6E,F*). Taken together our data support the model that the tight control of TORC1 activity by both GATOR1 and TSC is essential to the proper regulation of meiotic DSBs during *Drosophila* oogenesis.

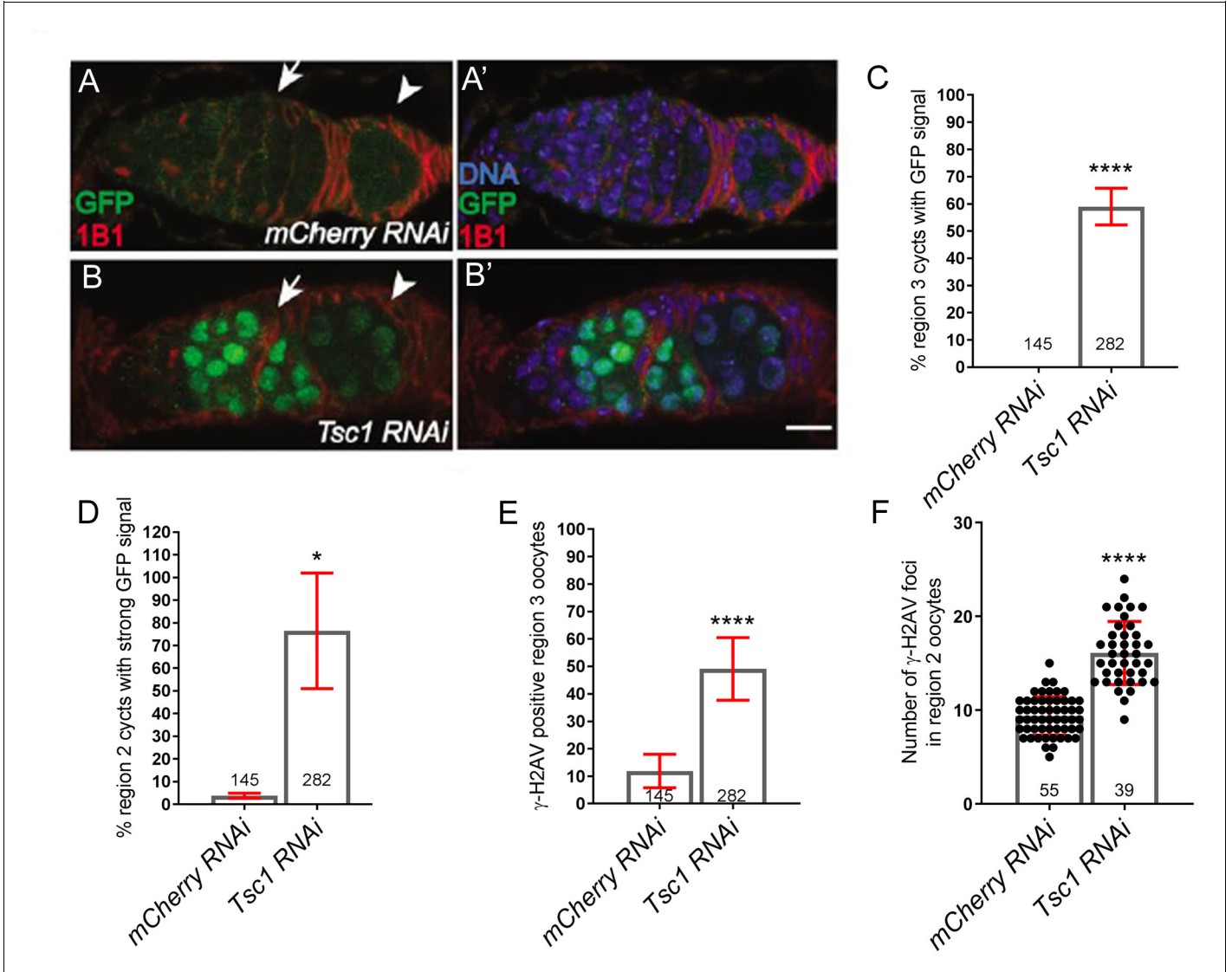

**Figure 6.** The TORC1 inhibitor TSC1 promotes genomic stability in early oocytes. Ovaries from (A) *p53R-GFP; MTD >mCherry RNAi* and (B) *p53R-GFP; MTD >Tsc1 RNAi* flies were stained for GFP (green) and 1B1 (red). In the *mCherry RNAi* (control) ovaries the p53-GFP expression is very low. In contrast, the *Tsc1 RNAi* ovaries have sustained GFP signal that persists into germarial region 3. Scale bars, 10 μm. (C) Percentage of germaria with strong p53R-GFP signal in region 2. (D) Percentage of germaria with sustained p53R-GFP signal in region 3. The γ-H2Av foci were determined in *MTD >mCherry RNAi* and *MTD >Tsc1 RNAi* ovaries. (E) Percentage of region three oocytes with γ-H2Av foci. (F) Quantification of γ-H2Av foci per oocyte in region 2a. Unpaired student T-test was used to calculate the statistical significance Error bars represent SD from at least three independent experiments. *p<0.05, ****p<0.0001.

DOI: https://doi.org/10.7554/eLife.42149.013

### *nprl3* mutant follicle cells are sensitive to genotoxic stress

In humans, mutations in components of TSC sensitize cells to multiple forms of genotoxic stress (*Paterson et al., 1982*; *Deschavanne and Fertil, 1996*; *Lee et al., 2007*; *Pai et al., 2016*). Thus, we predicted that the GATOR1 mutants, which have a two to three-fold increase in TORC1 activity, might have a globally diminished ability to respond to DNA damage. To test this model, we treated *nprl3* mutant larvae with the mutagen Methyl Methane Sulfonate (MMS) and compared the percentage of mutant animals that survived to adulthood relative to sibling heterozygous controls. MMS generates an array of DNA lesions including DSBs (*Magaña-Schwencke et al., 1982*). Notably, we found that *nprl3* mutant larvae were sensitive to DNA damage, with *nprl3/Df* transheterozygotes exhibiting a greater than 10-fold decrease in survival rates when exposed to 0.08% MMS (*Table 1*). These data support the idea that the GATOR1 complex plays a critical role in the response to genotoxic stress in both germline and somatic tissues.

Next, we wanted to determine if, as we observed in the female germline, mutations in GATOR1 components result in a delay in the repair of DSBs in somatic tissues. We exposed *Drosophila* females to 10 Gray (Gy) of γ-rays and then followed the dynamics of γ-H2Av staining in the somatically derived mitotically dividing follicle cells of the ovary. During *Drosophila* oogenesis, the somatic follicle cells divide mitotically until stage 6 of oogenesis at which point, they enter the endocycle. Prior to stage 6 of oogenesis wild-type and *nprl3/Df* follicle cells have very low levels of γ-H2Av staining (*Figure 7A,B*) (*Hong et al., 2007*). One hour after exposure to 10Gy of γ-H2Av both wild-type and *nprl3/Df* mutants have a dramatic increase in follicle cells with γ-H2Av foci (*Figure 7A',B'*). However, 6 hr after irradiation while wild-type somatic cells have decreased numbers of γ-H2Av positive follicle cells due to rapid DNA repair, *nprl3/Df* females retain elevated numbers of follicle cells with γ-H2Av foci. (*Figure 7A–C*). Thus, as is observed in the female germline, in *nprl3* mutants the somatic cells of the ovary exhibit a delay in the repair of DSBs.

### GATOR1 inhibits retrotransposon expression in *Drosophila*

Genotoxic stress, resulting from DNA damage, has been implicated in transposon activation in multiple organisms (*Bradshaw and McEntee, 1989*; *Walbot, 1992*; *Hagan et al., 2003*; *Beauregard et al., 2008*). These results are consistent with the model that genotoxic stress promotes TE activation (*McClintock, 1984*; *Harris et al., 2009*; *Wylie et al., 2016*). Therefore, we wanted to determine if retrotransposons expression is derepressed in the ovaries of GATOR1 mutants which exhibit several phenotypes consistent with increased genotoxic stress. Towards this end, we used qRT-PCR to compare expression levels for multiple retrotransposons in wild type versus *nprl2* and *nprl3* mutant ovaries. We found that *nprl2* and *nprl3* mutant ovaries have increased expression of multiple retrotransposons including TAHRE, Het-A, Indefix and Gypsy (*Figure 8A*). In contrast to *nprl2* and *nprl3* mutants, *Tsc1^RNAi* germline resulted in little or no increase in retrotransposon expression (*Figure 8—figure supplement 1*). From these results, we conclude that the

**Table 1.** *nprl2* and *nprl3* larvae are sensitive to the mutagen Methyl Methane Sulfonate.

| | MMS (0%) | | MMS (0.04%) | | MMS (0.08%) | |
|---|---|---|---|---|---|---|
| Genotype | % Obs. (# of Obs. / # of Total) | % of expected progeny | % Obs. (# of Obs. / # of Total) | % of expected progeny | % Obs. (# of Obs. / # of Total) | % of expected progeny |
| $nprl2^1/$ $nprl2^1$ | 27.6 (172/624) | **55.2%** (27.6/50) | 17.6 (99/561) | **35.2%** (17.6/50) | 13.7 (42/307) | **27.4%** (13.7/50) |
| $nprl3^1/Df$ | 27.1 (822/3038) | **81.4%** (27.1/33.3) | 8 (198/2472) | **24%** (8.0/33.3) | 2.4 (46/1925) | **7.2%** (2.4/33.3) |

Eclosion after exposure to the mutagen Methyl Methane Sulfonate. Third instar larvae derived from heterozygous parents were treated with the indicated concentration of MMS and the surviving adult progeny were scored. % Obs represents the total percentage of adults of the indicated mutant genotype divided by the total number of adults scored. % of expected progeny represents the percentage of mutant adults observed (% Obs) divided by the expected percentage of mutant adults based on the parental cross. For *nprl2*, which is on the X chromosome, the expected percentage of mutant progeny was 50%. For *nprl3*, which is on the 3rd chromosome, the expected percentage of mutant progeny was 33.3%. The observation that *nprl2* and *nprl3* have a lower percent survival than would be predicted at 0% MMS reflects the fact that these mutants are partially lethal in the absence of mutagen.

Obs.=Observed, # of Obs = Number of mutant adults scored, # of Total = Total number of adults scored.

DOI: https://doi.org/10.7554/eLife.42149.014

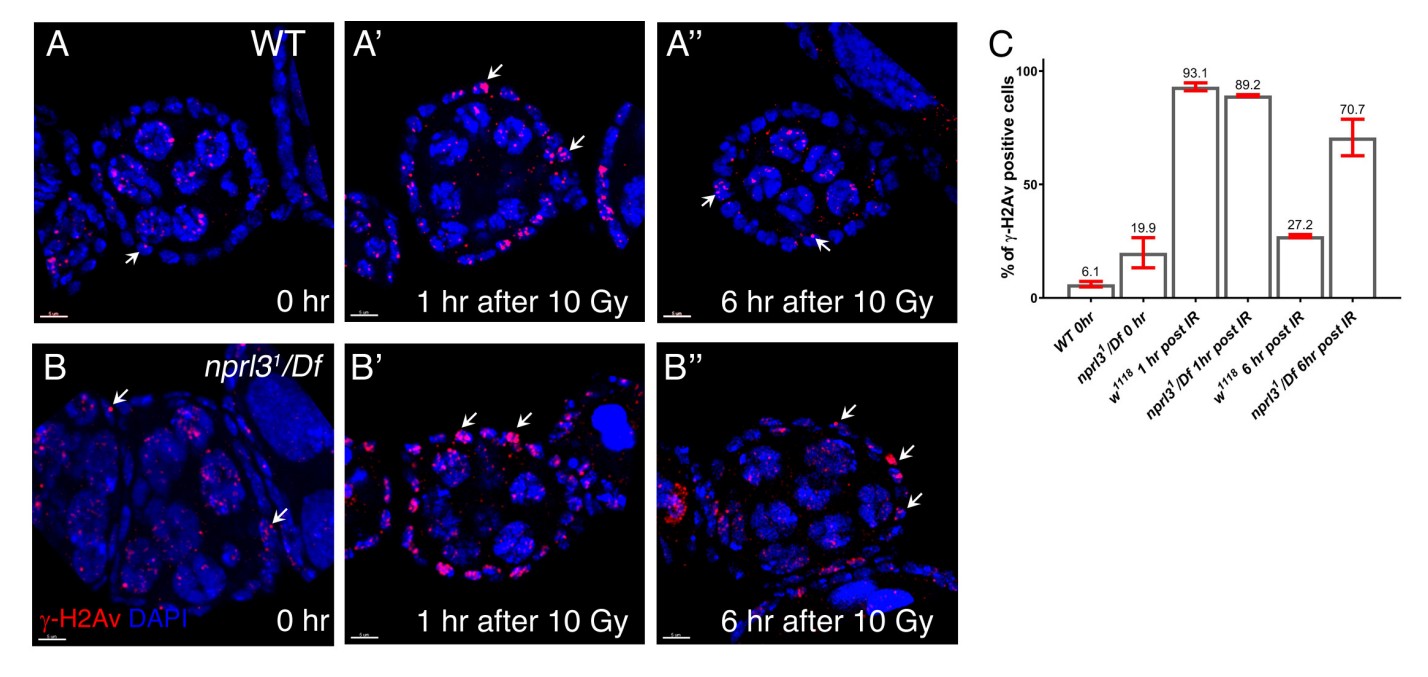

**Figure 7.** Nprl3 promotes DNA repair in somatically derived follicle cells. Egg chambers from fed (**A**) WT and (**B**) *nprl3¹/Df* females exposed to 10 Gy γ-irradiation. Ovaries were dissected at 0 hr (no irradiation) (**A, B**), 1 hr (**A', B'**) and 6 hr post-irradiation (**A'', B''**) and stained with antibodies against γ-H2Av (dsDNA breaks, Red), C(3)G (cyan) and the DNA dye DAPI (blue). (**C**) Quantification of γ-H2Av positive follicle cells from the indicated time points and genotypes. Note that an increased percentage of *nprl3/Df* follicle cells contain γ-H2Av foci 6 hr post irradiation relative to controls. Scale bar: 7 μm. Arrows denote γ-H2Av positive follicle cells.

DOI: https://doi.org/10.7554/eLife.42149.015

GATOR1 components *nprl2* and *nprl3* oppose retrotransposon expression in the female germline of *Drosophila*.

An additional possible connection between retrotransposon expression and the delayed repair of meiotic DSBs is suggested by our analysis of the DNA repair protein *spnA/Rad51*. As discussed above, *spnA* is a homolog of the DNA repair protein Rad51 which is conserved from yeast to humans (**Staeva-Vieira et al., 2003**). Rad51 catalyzes strand exchange between homologous DNA molecules and thus facilitates homologous recombination. In *Drosophila*, *spnA/Rad51* mutant females fail to repair meiotic DSBs (**Staeva-Vieira et al., 2003**). Intriguingly, we found that *spnA/Rad51* mutant ovaries had increased levels of retrotransposon expression (**Figure 8A**). Recently it has been reported that depletions of Rad51 result in the activation of Long interspersed repeat element 1 (LINE1) retrotransposons in HeLa cells (**Liu et al., 2018**). Thus, our data are consistent with the model that the increased levels of retrotransposon expression observed in GATOR1 mutant ovaries may be due, at least in part, to the delay in the repair of meiotic DSBs.

In *nprl3* mutant ovaries, the activation of p53 and the increase in γ-H2Av foci is dependent on the production of meiotic DSBs (**Figures 3** and **4**). To determine if the expression of retrotransposons in *nprl3* mutant ovaries also requires the meiotic DSB machinery, we examined *nprl3, mei-P22* double mutants. Using qRT-PCR we observed that retrotransposon expression was largely, but not completely, suppressed in *nprl3, mei-P22* double-mutant ovaries. Thus, meiotic DSBs trigger the expression of retrotransposons during *Drosophila* oogenesis in the *nprl3* mutant background (**Figure 8B**). However, it is important to note that these data suggest that the GATOR1 complex may also impact retrotransposon expression independent of meiotic DSBs as indicated by the relatively modest rescue of TAHRE over-expression observed in *nprl3, mei-P22* double mutants (**Figure 8B**).

Finally, we wanted to determine if the GATOR1 complex inhibits the activation of retrotransposons through the p53 pathway. To answer this question, we performed epistasis analysis by

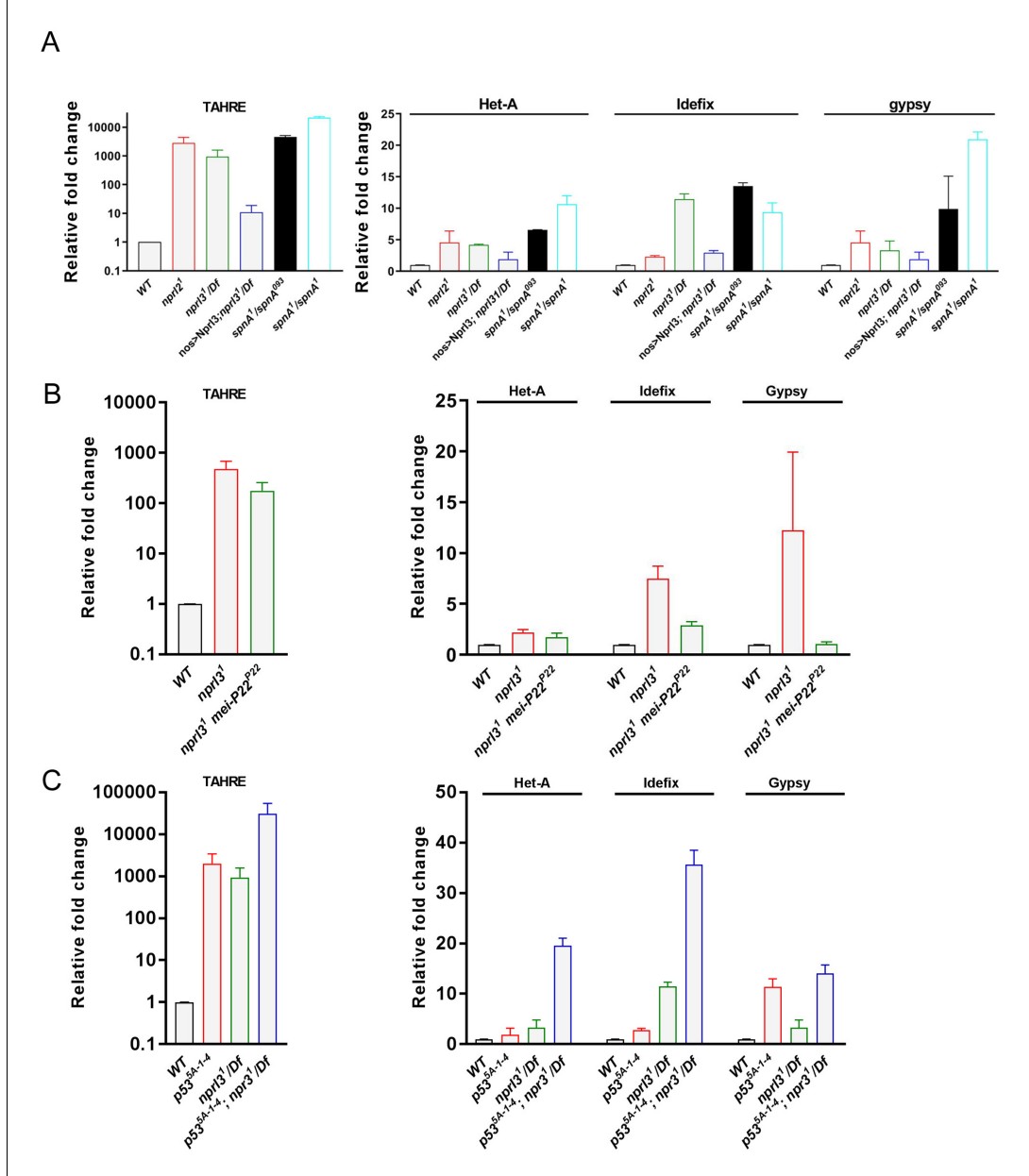

**Figure 8.** GATOR1 opposes retrotransposon expression in parallel to p53. (A) Quantitative RT-PCR analysis of expression levels for retrotransposons in wild type, *nprl2¹*, *nprl3¹/Df*, *nanos-GAL4; UAS-Nprl3; nprl3¹/Df*, *spnA¹/spnA⁰⁹³* and *spnA¹/spnA¹* ovaries. (B) Quantitative RT-PCR analysis of expression levels for transposons in wild type, *nprl3¹* and *mei-P22^P22^, nprl3¹* ovaries. (C) Quantitative RT-PCR analysis of expression levels for the transposons in wild type, *p53^5A-1-4^*, *nprl3¹/Df* and *nprl3¹/Df, p53^5A-1-4^* ovaries. Rp49 is used for normalization. Fold expression levels are relative to wild type. Error bars represent SD of three independent experiments.

DOI: https://doi.org/10.7554/eLife.42149.016

The following figure supplement is available for figure 8:

**Figure supplement 1.** Quantitative RT-PCR analysis of retrotransposon transcript levels in *Tsc1* knockdowns.

DOI: https://doi.org/10.7554/eLife.42149.017

generating double mutants that were homozygous for null alleles of both *p53* and *nprl3*. Strikingly, the *p53, nprl3* double mutant ovaries showed a dramatic increase in retrotransposon expression relative to either single mutant (*Figure 8C*). Thus, the *p53* and *nprl3* phenotypes are additive with respect to the inhibition of retrotransposon expression. These data strongly suggest that GATOR1

and p53 act through independent pathways to inhibit retrotransposon activation in the female germ-line during meiosis (*Figure 8C*).

## Discussion

Recent evidence implicates metabolic pathways as important regulators of meiotic progression and gametogenesis (*LaFever et al., 2010*; *Ferguson et al., 2012*; *Chi et al., 2016*; *Sieber et al., 2016*; *Guo et al., 2018*). Here we define a role for the GATOR complex, a conserved regulator of TORC1 activity, in the regulation of two events that impact germline genome stability: the response to meiotic DSBs and the inhibition of retrotransposon expression.

### The GATOR complex and the response to meiotic DSBs

We have previously shown that in *Drosophila,* mutations in the GATOR2 component *mio*, result in the constitutive activation of the GATOR1 pathway in the female germline but not in somatic tissues (*Iida and Lilly, 2004*; *Wei et al., 2014*). Here we demonstrate that the tissue specific requirement for *mio* during oogenesis is due, at least in part, to the generation of meiotic DBSs during oogenesis. In *Drosophila*, only the female germline undergoes meiotic recombination and thus experiences the genotoxic stress associated with developmentally programmed DSBs (*Hughes et al., 2018*). We show that in *mio* mutants, blocking the formation of meiotic DSBs prevents the constitutive downregulation of TORC1 activity thus allowing for the growth and development of the oocyte. These data are consistent with the model that meiotic DSBs trigger the activation of a TORC1 inhibitory pathway that must be opposed and/or attenuated by the GATOR2 component Mio (*Figure 9*).

While there are several possible models that might explain our data, we believe the most parsimonious explanation for our results is that the TORC1 inhibitory pathway activated by meiotic DSBs, involves both GATOR1 and TSC (*Figure 9*). This model is consistent with the ability of both GATOR1 and TSC depletions to rescue the *mio* mutant phenotype (*Wei et al., 2014*). Additionally, recent reports indicate that GATOR1 and TSC act in a common pathway to downregulate TORC1 activity in response to multiple upstream inhibitory inputs (*Demetriades et al., 2014*; *Menon et al., 2014*; *Demetriades et al., 2016*). Previously, we determined that in *Drosophila*, amino acid starvation induces a dramatic GATOR1/TSC dependent decrease in TORC1 activity in somatic tissues, that far exceeds any reduction in TORC1 activity observed in *GATOR2* null mutants (*Wei et al., 2014*; *Cai et al., 2016*). This observation strongly suggests that, in addition to the removal of the GATOR2 inhibition of GATOR1, there is an activation step that is required to fully potentiate the GATOR1/TSC pathway.

Thus, based on our data we propose the following model (*Figure 9*). Meiotic DSBs activate, or are required to maintain, a GATOR1/TSC dependent pathway that downregulates TORC1 activity in the female germline (*Figure 9A*). The GATOR2 component Mio is required to oppose or turnoff this pathway to prevent the constitutive downregulation of TORC1 activity in later stages of oogenesis. While we believe our data support the role of the GATOR1/TSC pathway, we concede that an alternative regulator of TORC1 activity may also be critical to the downregulation of TORC1 activity in response to meiotic DSBs.

### GATOR1 and TSC promote the repair of meiotic DSBs

Hyperactivation of TORC1 has been linked to defects in the DNA damage response in single celled and multicellular organisms (*Begley et al., 2004*; *Feng et al., 2007*; *Klermund et al., 2014*; *Pai et al., 2016*; *Ma et al., 2018*; *Xie et al., 2018*). The observation that meiotic DSBs likely promote the GATOR1 dependent downregulation of TORC1 activity during *Drosophila* oogenesis, suggested that limiting TORC1 activity may be important to the regulation of meiotic DSB repair. In our previous work, we found that GATOR1 mutant ovaries had TORC1 activity levels approximately three times higher than those observed in wild-type ovaries (*Wei et al., 2014*; *Cai et al., 2016*). Here we demonstrate that GATOR1 mutant ovaries exhibit multiple phenotypes consistent with the misregulation of meiotic DSB repair including, an increase in the steady state number of Mei-W68/Spo-11 induced DSBs, the retention of meiotic DSBs into later stages of oogenesis and the hyperactivation of p53. (*Cai et al., 2016*; *Wei et al., 2016*). Importantly, RNAi depletions of *Tsc1* partially phenocopied the GATOR1 ovarian defects. Thus, the misregulation of meiotic DSBs observed in

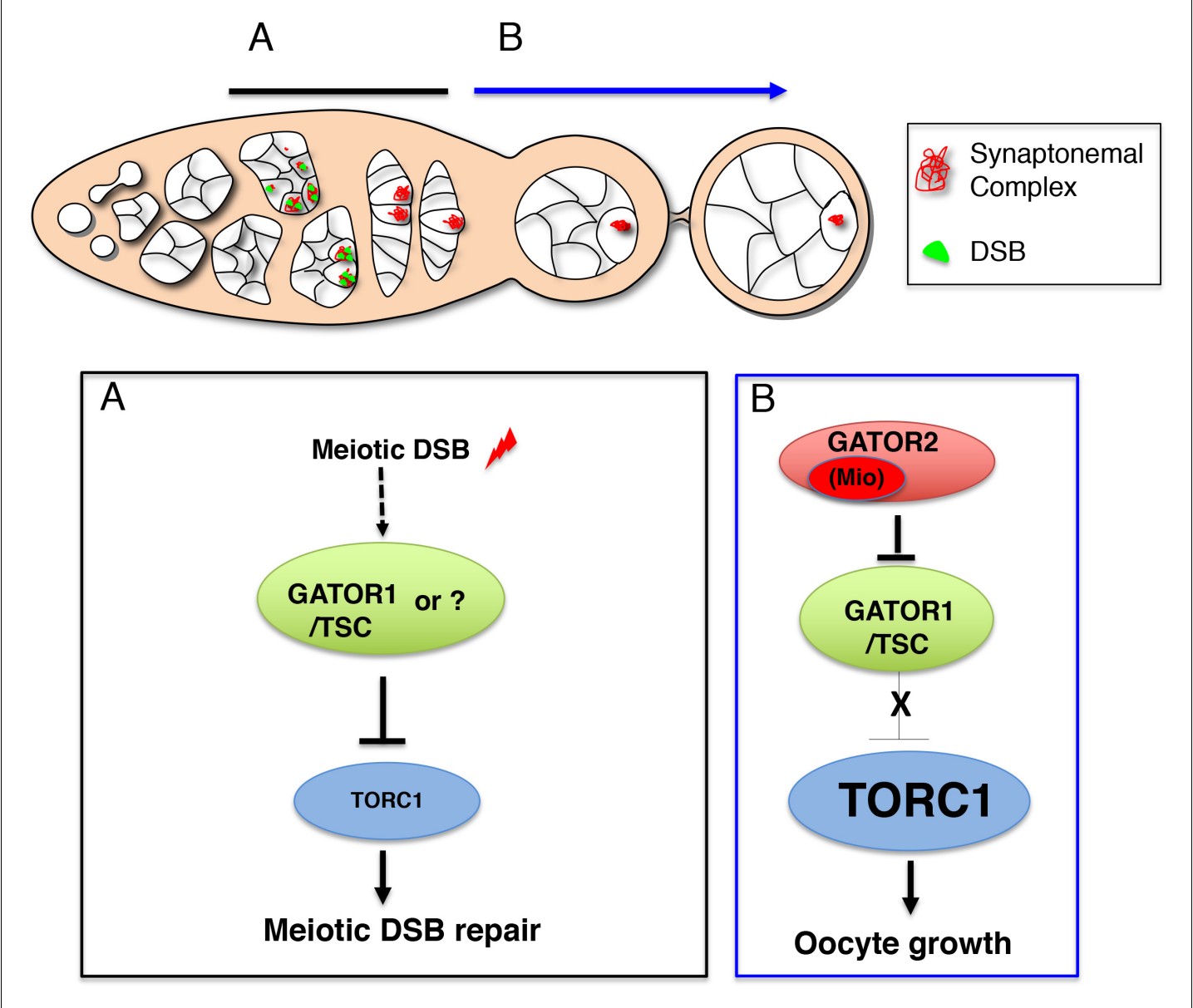

**Figure 9.** A working model for the role of the GATOR complex in the response to meiotic DSBs. (A) After ovarian cysts enter meiosis, meiotic DSBs function to activate and/or maintain a GATOR1/TSC dependent pathway to ensure low TORC1 activity in early prophase of meiosis I. Low TORC1 activity promotes the timely repair of meiotic DSBs. Currently, whether meiotic DSBs directly activate the GATOR1/TSC pathway or an alternative pathway that works in concert with, or in parallel to, GATOR1/TSC is not known. (B) Subsequently, the GATOR2 component Mio is required to attenuate the activity of the GATOR1/TSC pathway, thus allowing for increased TORC1 activity and the growth and development of the oocyte in later stages of oogenesis.

DOI: https://doi.org/10.7554/eLife.42149.018

GATOR1 mutant oocytes are due to high TORC1 activity and not to a TORC1 independent function of the GATOR1 complex.

Epistasis analysis between the GATOR1 component *nprl3* and the Rad51 homolog *spnA*, strongly suggest that GATOR1 impacts the repair, rather than the generation, of meiotic DSBs. We determined that double mutants of *nprl2* and the Rad51 homolog *spnA*, which is required for the repair of meiotic DSBs, have approximately the same number of DSBs as *spnA* single mutants. These data are consistent with GATOR1 and *spnA* influencing the common process of DNA repair and are inconsistent with GATOR1 mutants producing supernumerary breaks.

Our observations on the role of the GATOR1 complex during *Drosophila* oogenesis are particularly intriguing in light of similar meiotic defects observed in a *npr3* mutants in *Saccharomyces cerevisiae* (*Jordan et al., 2007*). In the sporulation proficient strain SK1, *npr3* mutant cells enter meiosis and express the transcription factor and master regulator of gametogenesis IME1 with wild-type kinetics (*Jordan et al., 2007*). Subsequently, *npr3* mutants exhibit a mild delay in the generation of meiotic DNA breaks, but a substantial delay in the repair of meiotic DSBs (*Jordan et al., 2007*). Thus, yeast and *Drosophila* SEACIT/GATOR1 mutants share a common meiotic phenotype, the delayed repair of meiotic DSBs. These results raise the intriguing possibility that low TORC1 activity may be a common feature of the early meiotic cycle in many organisms.

Notably, our data indicate that the delay in the repair of meiotic DSBs in GATOR1 mutants is due to the hyperactivation of the TORC1 downstream target S6K. S6K is a critical downstream effector of TORC1 that impacts multiple essential cellular processes including, but not limited to cell growth, energy balance and aging (*Magnuson et al., 2012*). Intriguingly, in mammals, S6K has been implicated in the regulation of the DNA damage response with hyperactivation of the TORC1-S6K pathway resulting in the accumulation of unrepaired DSBs and genome instability (*Lai et al., 2010*; *Xie et al., 2018*). Thus, similar to what is reported in mammals, our data are consistent with the model that the hyperactivity of the TORC1/S6K axis delays the repair of DSBs in *Drosophila*.

Finally, we determined that GATOR1 mutants have a diminished response to DSBs outside the female germline in somatic tissues of *Drosophila*. Similar to what is observed in TSC mutant cells in humans that have increased levels of TORC1 activity, we find that GATOR1 mutant embryos have a reduced ability to survive low levels of γ-irradiation (*Paterson et al., 1982*; *Deschavanne and Fertil, 1996*; *Lee et al., 2007*; *Pai et al., 2016*). Moreover, in the somatic follicle cells of the ovary we observed a delay in the repair of DSBs after adult females are exposed to low levels of γ-irradiation. Thus, in *Drosophila* inappropriately high TORC1 activity delays the repair of DSBs in both the germline and somatic tissues.

## GATOR1 opposes retrotransposon expression

The initiation of homologous recombination through the programmed generation of DNA double-stranded breaks (DSBs) is a universal feature of meiosis (*McKim and Hayashi-Hagihara, 1998*; *Gray and Cohen, 2016*). DSBs represent a dangerous form of DNA damage that can result in dramatic and permanent changes to the germline genome (*Alexander et al., 2010*). To minimize this destructive potential, the generation and repair of meiotic DSBs is tightly controlled in space and time (*Longhese et al., 2009*). The activation of transposable elements represents an additional threat to genome integrity in germ line cells (*Crichton et al., 2014*; *Toth et al., 2016*). Genotoxic stress, resulting from DNA damage, has been implicated in the deregulation of transposons in multiple organisms (*Bradshaw and McEntee, 1989*; *Walbot, 1992*; *Hagan et al., 2003*; *Beauregard et al., 2008*). Thus, germ line cells may be at an increased risk for transposon derepression due to the genotoxic stress associated with meiotic recombination. Consistent with this hypothesis, germ line cells have evolved extensive surveillance systems to detect and silence transposons beyond the pathways present in most somatic tissues (*Khurana and Theurkauf, 2010*; *Ku and Lin, 2014*; *Toth et al., 2016*).

Previous studies have shown that DNA damage promotes the deregulation of retrotransposon in multiple organisms, including *Drosophila* (*McClintock, 1984*; *Bradshaw and McEntee, 1989*; *Walbot, 1992*; *Hagan et al., 2003*; *Beauregard et al., 2008*; *Molla-Herman et al., 2015*; *Wylie et al., 2016*). In line with these studies, we find that in GATOR1 mutants, the DSBs that initiate meiotic recombination trigger the deregulation of retrotransposon expression. Similarly, *p53* mutant females derepress retrotransposon expression during oogenesis, but as observed in GATOR1 mutants, primarily in the presence of meiotic DSBs (*Wylie et al., 2016*). Double mutants of *nprl3*, *p53* exhibit a dramatic increase in retrotransposon expression relative to either *p53* or *nprl3* single mutants, implying that *p53* and GATOR1 act through independent pathways to repress retrotransposon expression in the female germline. One possibility is that both GATOR1 and p53 independently impact genome stability. Thus, disabling both pathways may have an additive effect on both genome stability and retrotransposon expression. Consistent with the hypothesis that genome instability drives retrotransposon expression, we find that mutants in *spnA/Rad51*, which fail to repair meiotic DSBs, also exhibit increased transcription of multiple retrotransposons. Intriguingly, the SpnA homolog Rad51, as well as other genes required for DNA repair, was recently identified in a high throughput screen for

genes that suppress (Long Interspersed Element-1) LINE1 expression in mammalian tissue culture cells (*Liu et al., 2018*).

However, our data also suggest that the GATOR1 complex may influence retrotransposon expression independent of the regulation of TORC1 activity. While both GATOR1 and TSC are required for the efficient repair of meiotic DSBs, in contrast to GATOR1 mutant ovaries, we observed little to no increase in retrotransposon expression in the *Tsc1* depleted ovaries. We believe reflects the incomplete depletion of Tsc1 by RNAi resulting in a reduced retention of meiotic DSBs relative to GATOR1 mutants (*Figure 6*). However, a second possibility is that the GATOR1 complex inhibits retrotransposon expression independent of TORC1 inhibition. As is observed with *spnA* the depletion of GATOR1 components, but not TSC components result in the activation of LINE1 expression in HeLa cells (*Liu et al., 2018*). Taken together, these data hint that the GATOR1 complex may impact retrotransposon expression in the germline via two independent pathways: First by promoting the repair of meiotic DSBs through the downregulation of TORC1 activity and second via a pathway that functions independent of TORC1 inhibition.

Genes encoding components of the GATOR1 complex are often deleted in cancers (*Lerman and Minna, 2000*; *Ji et al., 2002*; *Ueda et al., 2006*; *Bar-Peled et al., 2013*). As is observed in GATOR1 mutants, cancer cells frequently have increased TORC1 activity, increased genomic instability and increased retrotransposon expression. Thus, in the future it will be important to identify the molecular mechanism by which the GATOR1 complex influences both the response to genotoxic stress and the expression of retrotransposons under both normal and pathological conditions.

# Materials and methods

## Key resources table

| Reagent type or resource | Designation | Source or reference | Identifiers | Additional information |
|---|---|---|---|---|
| Gene (*Drosophila melanogaster*) | Atg1 | FlyBase | FBgn0260945 | |
| Gene (*Drosophila melanogaster*) | Iml1 | FlyBase | FBgn0035227 | |
| Gene (*Drosophila melanogaster*) | Loki | FlyBase | FBgn0019686 | |
| Gene (*Drosophila melanogaster*) | Mio | FlyBase | FBgn0031399 | |
| Gene (*Drosophila melanogaster*) | Mei-W68 | FlyBase | FBgn0002716 | |
| Gene (*Drosophila melanogaster*) | Mei-P22 | FlyBase | FBgn0016036 | |
| Gene (*Drosophila melanogaster*) | Nprl2 | FlyBase | FBgn0030800 | |
| Gene (*Drosophila melanogaster*) | Nprl3 | FlyBase | FBgn0036397 | |
| Gene (*Drosophila melanogaster*) | p53 | FlyBase | FBgn0039044 | |
| Gene (*Drosophila melanogaster*) | Spn-A | FlyBase | FBgn0003479 | |

*Continued on next page*

Continued

| Reagent type or resource | Designation | Source or reference | Identifiers | Additional information |
|---|---|---|---|---|
| Gene (*Drosophila melanogaster*) | S6K | FlyBase | FBgn0283472 | |
| Gene (*Drosophila melanogaster*) | Thor | FlyBase | FBgn0261560 | |
| Gene (*Drosophila melanogaster*) | Tsc1 | FlyBase | FBgn0026317 | |
| Genetic reagent (*Drosophila melanogaster*) | Atg1$^{\Delta 3D}$, FRT 2A/Tm3, sb1 | FlyBase, PMID: 24098761 | FBal0176392 | |
| Genetic reagent (*Drosophila melanogaster*) | iml1$^1$ | FlyBase, PMID: 27672113 | FBal0325028 | FlyBase symbol: iml1$^1$ |
| Genetic reagent (*Drosophila melanogaster*) | loki$^{P6}$ | FlyBase, PMID: 14729967 | FBal0216721 | FlyBase symbol: lok$^{P6}$ |
| Genetic reagent (*Drosophila melanogaster*) | mio$^2$ | FlyBase, PMID: 14973288 | FBal0158954 | FlyBase symbol: mio$^2$ |
| Genetic reagent (*Drosophila melanogaster*) | mei-P22$^{P22}$ | Bloomington Drosophila Stock Center | BDSC:4931 | Genotype: y$^1$ w$^1$/Dp(1;Y)y$^+$; mei-P22$^{P22}$; sv$^{spa-pol}$ |
| Genetic reagent (*Drosophila melanogaster*) | mei-W68$^1$ | Bloomington Drosophila Stock Center | BDSC:4962 | FlyBase symbol: mei-W86$^1$ |
| Genetic reagent (*Drosophila melanogaster*) | nprl2$^1$ | FlyBase, PMID: 27672113 | FBal0325026 | FlyBase symbol: nprl2$^1$ |
| Genetic reagent (*Drosophila melanogaster*) | nprl3$^1$ | FlyBase, PMID: 27166823 | FBal0319815 | FlyBase symbol: nprl3$^1$ |
| Genetic reagent (*Drosophila melanogaster*) | p53$^{5A-1-4}$ | Bloomington Drosophila Stock Center | BDSC:6815 | Genotype: y$^1$ w$^{1118}$; p53$^{5A-1-4}$ |
| Genetic reagent (*Drosophila melanogaster*) | S6K$^{l-1}$ | Bloomington Drosophila Stock Center | BDSC:32552 | Genotype: y[1] w[*]; S6k[l-1]/TM6B, P{y[+t7.7] ry[+t7.2]=Car20y}TPN1, Tb[1] |
| Genetic reagent (*Drosophila melanogaster*) | spnA$^1$ | Bloomington Drosophila Stock Center | BDSC:3322 | Genotype: Dp(1;Y)B$^S$; ru$^1$ st$^1$ e$^1$ spn-A$^1$ ca$^1$/ TM3, Sb$^1$ |
| Genetic reagent (*Drosophila melanogaster*) | spnA$^{093A}$ | FlyBase, PMID: 14592983 | FBal0151428 | FlyBase symbol: spn-A$^{093A}$ |
| Genetic reagent (*Drosophila melanogaster*) | thor$^2$ | Bloomington Drosophila Stock Center | BDSC:9559 | Genotype: y[1] w[*]; Thor[2] |
| Genetic reagent (*Drosophila melanogaster*) | Df(3L)ED4238 | Bloomington Drosophila Stock Center | BDSC:8052 | Genotype: w$^{1118}$; Df(3L)ED4238, P{3'.RS5+3.3'}ED4238/ TM6C, cu$^1$ Sb$^1$ |
| Genetic reagent (*Drosophila melanogaster*) | Df(3L)ED4515 | Bloomington Drosophila Stock Center | BDSC:9071 | Genotype: w$^{1118}$; Df(3L)ED4515, P{3'.RS5+3.3'}ED4515/ TM6C, cu$^1$ Sb$^1$ |

*Continued on next page*

Continued

| Reagent type or resource | Designation | Source or reference | Identifiers | Additional information |
|---|---|---|---|---|
| Genetic reagent (*Drosophila melanogaster*) | nanos-Gal4 | FlyBase, PMID: 9501989 | FBrf0100715 | FlyBase symbol: GAL4[VP16.nos.UTR] |
| Genetic reagent (*Drosophila melanogaster*) | p53R-GFP | FlyBase, PMID: 20522776 | FBrf0210965 | FlyBase symbol: GFP[rpr.p53R.Tag:NLS(Unk)] |
| Genetic reagent (*Drosophila melanogaster*) | MTD-Gal4 | Bloomington Drosophila Stock Center | BDSC:31777 | Genotype: P{otu-GAL4::VP16.R}1, w[*]; P{GAL4-nos.NGT}40; P{GAL4::VP16-nos. UTR}CG6325[MVD1] |
| Genetic reagent (*Drosophila melanogaster*) | Tsc1 RNAi | Bloomington Drosophila Stock Center | BDSC:35144 | Genotype: y[1] sc[*] v[1]; P{TRiP.GL00012}attP2 |
| Genetic reagent (*Drosophila melanogaster*) | Iml1 RNAi | Bloomington Drosophila Stock Center | BDSC:57492 | Genotype: y[1] sc[*] v[1] sev[21]; P{y[+t7.7] v[+t1.8]=TRiP. HMC04806} attP40 |
| Genetic reagent (*Drosophila melanogaster*) | mCherry RNAi | Bloomington Drosophila Stock Center | BDSC:35787 | Genotype: y[1] sc[*] v[1] sev[21]; P{y[+t7.7] v[+t1.8]=UAS mCherry. VALIUM10}attP2 |
| Genetic reagent (*Drosophila melanogaster*) | UAS-Nprl3 | FlyBase, PMID: 27672113 | FBrf0234182 | FlyBase symbol: Nprl3[UASp.Tag:FLAG,Tag:HA] |
| Genetic reagent (*Drosophila melanogaster*) | FRT80B | Bloomington Drosophila Stock Center | BDSC:1988 | Genotype: w[*]; P{ry[+t7.2]=neoFRT} 80B ry[506] |
| Genetic reagent (*Drosophila melanogaster*) | FRT2A | Bloomington Drosophila Stock Center | BDSC:1997 | Genotype: w[*]; P{w[+mW.hs]=FRT(w[hs])}2A |
| Genetic reagent (*Drosophila melanogaster*) | FRT2A, ubi-GFPnls | Bloomington Drosophila Stock Center | BDSC: 5825 | Genotype: w[1118]; P{w[+mC]=Ubi GFP.nls}3L1 P{Ubi-GFP.nls}3L2 P {w[+mW.hs]=FRT(w[hs])}2A |
| Genetic reagent (*Drosophila melanogaster*) | hsFLP; FRT80B, lacZ | Bloomington Drosophila Stock Center | BDSC:6341 | Genotype: P{hsFLP}22, y[1] w[*]; P{arm-lacZ.V} 70 C P{neoFRT}80B |
| Genetic reagent (*Drosophila melanogaster*) | UAS-mCherry RNAi | Bloomington Drosophila Stock Center | BDSC:35785 | Genotype: y[1] sc[*] v[1]; P{VALIUM20-mCherry}attP2 |
| Genetic reagent (*Drosophila melanogaster*) | nos-Flp | This paper. | | Lilly Lab. |
| Antibody | Goat anti-Rabbit, mouse Alexa 488–568-594–647- secondaries | Thermo Fisher | | Immunofluorescence (1:1000) |
| Antibody | anti-dS6K (Guinea pig polyclonal) | PMID: 20444422 | | Western Blot (1:5000) |
| Antibody | anti-phospho-Thr398-S6K (Rabbit polyclonal) | Cell Signaling Technologies | 9209; RRID:AB_2269804 | Western Blot (1:1000) |
| Antibody | anti-GFP (Rabbit polyclonal) | Invitrogen | A11122; RRID:AB_221569 | Immunofluorescence (1:1000) |
| Antibody | anti-g-H2Av (Rabbit poly clonal) | Active Motif | 39117; RRID:AB_2793161 | Immunofluorescence (1:1000) |

*Continued on next page*

Continued

| Reagent type or resource | Designation | Source or reference | Identifiers | Additional information |
|---|---|---|---|---|
| Antibody | anti-C(3)G (clone 1A8 and 1G2) (Mouse monoclonal) | PMID: 15767569 | | Immunofluorescence (1:200) |
| Antibody | anti-C(3)G (Rabbit polyclonal) | PMID: 12588841 | | Immunofluorescence (1:3000) |
| Antibody | anti-1B1 (Mouse monoclonal) | Developmental Studies Hybridoma Bank | 1B1; RRID:AB_528070 | Immunofluorescence (1:100) |
| Antibody | anti-g-H2Av (Mouse monoclonal) | Developmental Studies Hybridoma Bank | UNC93-5.2.1; RRID:AB_2618077 | Immunofluorescence (1:5000) |
| Antibody | anti-Phospho-4E-BP1 (Thr37/46) (236B4) (Rabbit monoclonal) | Cell Signaling Technologies | 2855; RRID:AB_560835 | Immunofluorescence (1:200) |
| Commercial assay or kit | ECL | PerkinElmer | NEL105001EA | |
| Chemical compound, drug | methyl methanesulfonate (MMS) | Sigma | 129925–5G | |
| Commercial assay or kit | RNAeasy Kit | Qiagen | 74104 | |
| Commercial assay or kit | cDNA Reverse Transcription Kit | Thermo Fischer | 11752 | |
| Commercial assay or kit | Power SYBR green mastermix | Thermo Fischer | A25742 | |

## Fly stocks

All fly stocks were maintained at 25°C on standard media. The *p53R-GFP* transgenic line was a gift from John M. Abrams (*Lu et al., 2010*). The germline specific driver nanos-Gal4 was obtained from Ruth Lehmann (*Van Doren et al., 1998*). The *spnA$^{093A}$* stock was a gift from Ruth Lehmann (*Staeva-Vieira et al., 2003*). The *nprl2$^1$*, *nprl3$^1$*, *iml1$^1$*, and *UAS-Nprl3* were described previously (*Cai et al., 2016*; *Wei et al., 2016*). The stocks *w$^{1118}$; Df(3L)ED4515, P{3'.RS5+3.3'}ED4515/TM6C, cu$^1$ Sb$^1$* (BDSC#9071), *w$^{1118}$; Df(3L)ED4238, P{3'.RS5+3.3'}ED4238/TM6C, cu$^1$ Sb$^1$* (BDSC#8052), *w$^{1118}$; P{neoFRT}82B P{Ubi-mRFP.nls}3R* (BDSC#30555), *P{hsFLP}22, y$^1$ w$^*$; P{arm-lacZ.V}70 C P{neoFRT}80B* (BDSC#6341), *MTD-GAL4 (P{w[+mC]=otu-GAL4::VP16.R}1, w[*] P{w[+mC]=GAL4 nos.NGT}40; P{w[+mC]=GAL4::VP16 nos.UTR}CG6325[MVD1], BDSC#31777)*, UAS-Tsc1 RNAi (*y$^1$ sc$^*$ v$^1$; P{TRiP.GL00012}attP2, BDSC#35144*), UAS-mCherry RNAi (*y$^1$ sc$^*$ v$^1$; P{VALIUM20-mCherry}attP2*), *y$^1$ w$^{1118}$; p53$^{5A-1-4}$* (BDSC#6815), *y$^1$ w$^1$/Dp(1;Y)y$^+$; mei-P22$^{P22}$; svspa-pol* (BDSC#4931), *mei-W68$^1$* (BDSC#4932) and *Dp(1;Y)B$^S$; ru$^1$ st$^1$ e$^1$ spn-A$^1$ ca$^1$/TM3, Sb$^1$* (BDSC#3322) were obtained from Bloomington Stock Center.

## Western blot analysis

The protocol was adapted from *Bjedov et al. (2010)*. Briefly 6 pairs of ovaries were freshly dissected in cell insect media and homogenized in 30 µl of 4x Laemmli loading sample buffer (Invitrogen, #NP0008) containing 10x sample reducing agent (Invitrogen, #NP009). Extracts were cleared by centrifugation and boiled for 10 min at 90°C. 10 µl of protein extract was loaded per lane on polyacrylamide gel (Invitrogen, #NP0335). Proteins were separated and transferred to nitrocellulose membrane. Primary antibodies used were as follows: guinea pig anti-dS6K (gift of Aurelius Teleman,1:5,000,) (*Hahn et al., 2010*) and rabbit anti-phospho-Thr398-S6K (Cell Signaling Technologies #9209, 1:1,000). HRP- conjugated secondary antibodies (Jackson Immunoresearch, AffiniPure anti-rabbit #111-005-144 and anti-guinea pig #106-005-003) were used. Blots were developed using the ECL detection system (PerkinElmer, #NEL105001EA). Western blots were analyzed using ImageJ program (US National Institutes of Health).

## Immunofluorescence and microscopy

Immunofluorescence was performed as previously described (*Hong et al., 2003*; *Iida and Lilly, 2004*). Primary antibodies used were as follows: rabbit anti-GFP (Invitrogen, 1:1000); rabbit anti-γ—H2Av (Active Motif, 1:500); rabbit anti-C(3)G 1:3000 (*Hong et al., 2003*); mouse anti-1B1 (Developmental Studies Hybridoma Bank, 1:100); mouse anti-γ-H2Av (Developmental Studies Hybridoma Bank, 1:5000); mouse anti-C(3)G (kindly provided by R. Scott Hawley, 1:200) (*Page and Hawley, 2001*). Alexa-488 and Alex-594 (Invitrogen, 1:1000) secondary antibodies were used for fluorescence. After staining, ovaries were mounted in prolong gold antifade reagent with DAPI (Life Technology). Images were acquired on either a Leica SP5 confocal microscope or Zeiss LSM 880 with Airyscan confocal microscope.

## γ-H2Av foci quantification

To score the number of γ-H2Av foci per oocyte, ovaries were stained with antibodies against C(3)G and γ-H2Av as well as the DNA dye DAPI. Pro-oocytes and oocytes were identified by the pattern of anti-C(3)G staining (*Page and Hawley, 2001*). Multiple Z-sections encompassing an entire region 2a pro-oocyte nucleus were acquired by Leica SP5 confocal microscope or Zeiss LSM 880 with Airyscan confocal microscopy. The obtained z-stacks of images were deconvolved to remove out-of-focus light and z-distortion with Huygens Professional software (Scientific Volume Imaging) and clearly defined γ-H2Av were counted manually in the ovarian cyst with the highest levels of γ-H2Av staining. Alternatively, 3D images were rendered by using Imaris software (Bitplane) and a graph workstation equipped with NVIDIA Quadro 3D vision system. Clearly defined γ-H2Av foci were visualized and counted by using Imaris spots module or ImageJ to define the total number of foci per nucleus.

## MMS sensitivity assay

The assay was performed as described in *Ghabrial et al. (1998)*. Briefly, 10 males and 10 virgin females were mated in vials for 2 days at 25°C. Parents were transferred into German food (Genesee Scientific, Cat#66–115, Day 1) vials for 24 hr at 25°C and allowed to lay eggs. On Day two the parents were removed, and eggs were allowed to mature for 24 hr. Subsequently, the first and second instar larvae were treated with 250 μL of either 0.04% or 0.08% of the mutagen methyl methane sulfonate (MMS) (Sigma, Cat#129925–5G). Control larvae were treated with 250 μL water. After eclosion, the number of heterozygous and homozygous mutant flies were determined, and the percentage of each genotype was calculated.

## Gamma irradiation assay

Wild type and mutant flies were fed wet yeast for two days and vials containing flies were exposed to 10Gy γ-IR in a Mark-1 γ-irradiator (JL Shepherd and Associates, San Fernando, CA). After irradiation, flies were incubated at 25°C and ovaries were collected for immunostaining assays at indicated time points.

## Retrotransposon expression analysis

Total RNA was isolated from dissected ovaries using the RNeasy Kit (Qiagen) and treated with DNase. cDNA was generated using High-capacity cDNA Reverse Transcription Kit (Thermo Fisher). Real-time PCR was performed with Power SYBR green Mastermix (Thermo Fisher) using the following primers:

Rp49 Forward: CCGCTTCAAGGGACAGTATC;
Rp49 Reverse: GACAATCTCCTTGCGCTTCT;
TAHRE Forward: CTGTTGCACAAAGCCAAGAA;
TAHRE Reverse: GTTGGTAATGTTCGCGTCCT;
Het-A Forward: TCCAACTTTGTAACTCCCAGC;
Het-A Reverse: TTCTGGCTTTGGATTCCTCG;
Idefix Forward: TGAAGAAAAGAAGGGCGAGA;
Idefix Reverse: TTCTGCTGTTGATGCTTTGG;
Gypsy Forward: CCAGGTCGGGCTGTTATAGG;
Gypsy Reverse: GAACCGGTGTACTCAAGAGC.

The rp49 was used for normalization.

## Acknowledgements

Multiple stocks used in this study were obtained from the Bloomington *Drosophila* Stock Center (supported by NIH Grant P40OD018537). Research reported in this publication was supported by the *Eunice Kennedy Shriver* National Institute of Child Health and Human Development Intramural Research program (to MAL, HD00163 16), the Natural Science Foundation of Jiangsu Province (to YW, BK20181456) and the National Natural Science Foundation of China (to YW, 31872287).

## Additional information

### Funding

| Funder | Grant reference number | Author |
|---|---|---|
| Eunice Kennedy Shriver National Institute of Child Health and Human Development | Intramural Program HD00163 16 | Mary A Lilly |
| National Science Foundation of Jiangsu Province | BK20181456 | Youheng Wei |
| National Natural Science Foundation of China | 31872287 | Youheng Wei |

The funders had no role in study design, data collection and interpretation, or the decision to submit the work for publication.

### Author contributions

Youheng Wei, Conceptualization, Data curation, Formal analysis, Validation, Investigation, Methodology, Writing—review and editing; Lucia Bettedi, Data curation, Formal analysis, Validation, Investigation, Writing—review and editing; Chun-Yuan Ting, Data curation, Formal analysis, Validation, Investigation, Methodology; Kuikwon Kim, Data curation, Formal analysis, Investigation, Methodology; Yingbiao Zhang, Formal analysis, Validation, Investigation, Methodology; Jiadong Cai, Investigation; Mary A Lilly, Conceptualization, Resources, Formal analysis, Funding acquisition, Investigation, Writing—original draft, Project administration, Writing—review and editing

### Author ORCIDs

Chun-Yuan Ting http://orcid.org/0000-0002-2302-4203
Mary A Lilly https://orcid.org/0000-0003-1564-619X

### Decision letter and Author response

Decision letter https://doi.org/10.7554/eLife.42149.021
Author response https://doi.org/10.7554/eLife.42149.022

## Additional files

### Supplementary files

• Transparent reporting form DOI: https://doi.org/10.7554/eLife.42149.019

### Data availability

All data generated or analysed during this study are included in the manuscript and supporting files.

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
