## [Decision Letter]

Thank you for submitting your article "The GATOR complex regulates an essential response to meiotic double-stranded breaks in *Drosophila*" for consideration by *eLife*. Your article has been reviewed by three peer reviewers, and the evaluation has been overseen by a Reviewing Editor and Jessica Tyler as the Senior Editor. The reviewers have opted to remain anonymous.

I have appended the reviews in their entirety below. As you can see, the reviews are reasonably well aligned, and the reviewers were favorable overall about your interesting study. However they have requested significant revisions that we hope can clarify some outstanding issues. These requests seem reasonable and I suggest you do your best to address those that are technically straightforward within a reasonable timeframe. While I agree with the reviewers that looking further into the GATOR dependency, determining whether DSBs in WT inhibit TOR, and how DSBs mechanistically affect TOR (is it via S6K?) are important directions, delineating a complete, precise mechanism is not necessarily essential for this paper to be published in *eLife*, so long as the conclusions are clear and warranted.

Reviewer #1:

In this manuscript, Wei and colleagues delineate a pathway whereby double-stranded breaks (DSBs) inhibit mTORC1 activity via the GATOR1 complex, and this in turn allows repair of the double-stranded breaks. Overall, this is an interesting finding. A couple of the causality links of this story, however, need to be further tested to make the story solid:

1) Figure 1C shows that inhibition of DSB formation leads to increased mTORC1 activity in a mutant condition where mTORC1 activity is aberrantly low due to loss-of-function of GATOR2 (*mio^2^* mutation). However, this does not show that in the wildtype condition DSBs affect mTOR activity. If the model is correct, the DSBs that occur normally in the ovary should be inhibiting mTORC1. Removing these DSBs should alleviate this suppression. So does the *mei-W68^1^* mutation increase S6K phosphorylation in the ovary compared to wildtype animals?

2) The role of GATOR1 in mediating the effect of DSBs on mTORC1 needs to be tested more rigorously using loss-of-function epistasis. From the data in Figure 1C, the authors conclude DSBs affect mTORC1 via GATOR1. However, the data do not exclude the alternate possibility that DSBs and GATOR1 affect mTORC1 via parallel pathways. One way to test this is to assay S6K phosphorylation in ovaries of the following genotypes:

- WT

- *mei-W68^1^*

- *iml1*- (GATOR1 loss-of-function)

- *iml1-, mei-W69^1^* double mutants

In addition, this manuscript is lacking molecular mechanisms. How does mTORC1 activity affect DSB repair? Is it through delaying cell cycle progression? Is it through affecting transcription or translation of proteins required for repair? Alternatively, the upstream mechanism how DSBs affect GATOR1 activity is also missing. This may be a significant amount of work. Hence, it is an editorial decision how much molecular mechanism is wished in the present manuscript.

Reviewer #2:

Wei and coworkers describe some interesting connections between TOR signaling, meiotic double strand breaks and genotoxic stress in the *Drosophila* ovary. This work builds on their previous characterization of the TORC1 regulators GATOR1 and GATOR2, the latter of which was shown to have germline-specific functions. Here they demonstrate that TORC1 activity is sensitive to DSB formation in the context of Mio/GATOR2 mutant ovaries, providing a possible explanation for these germline-specific effects. Under conditions of high TORC1 activity in GATOR1 mutants, resolution of meiotic DSBs is delayed, leading to increases in p53 activity, DNA damage sensitivity, and retrotransposon expression. Interestingly, some but not all of these effects are phenocopied by germline depletion of *Tsc1*, indicating some possible TORC1-independent functions of GATOR1.

This work is timely and well described, and adds to a growing literature examining the role of metabolism in gametogenesis. Overall the data are lovely and appear well controlled. Additional insight into the mechanisms behind these novel observations would strengthen the manuscript, and experiments that probe these issues in somatic cells would make it of more general interest to *eLife* readers.

1) The authors demonstrate that in *mio* mutants, disruption of DSB formation leads to increased TORC1 activity, and they repeatedly describe this result as a GATOR-dependent response. However, the observation that TORC1 activity responds to the presence or absence of DSBs in ovaries lacking *mio* demonstrates that GATOR2 is not required for this response (in contrast to the model depicted in Figure 6), and would suggest that it is likely independent of GATOR1 as well. The role of GATOR1 is not directly tested here, and DSBs could in principle affect TORC1 activity through any one of multiple positive or negative TORC1 regulators/components. Without evidence that GATOR1 is specifically/selectively involved in this signal, I don't see the justification for calling this a GATOR1-dependent effect in the Title, Abstract, etc.

2) The requirement for other well characterized TORC1 inputs (e.g., nutrient and insulin pathways) should be addressed. A more thorough probing of the mechanism by which DSBs affect TORC1 activity should be included here, and the effects of *mei-P22* mutation on TORC1 activity should be examined in otherwise wildtype animals and other mutations/conditions that lead to low TORC1 signaling.

3) The proximal signals linking DSBs to TORC1 downregulation also should be addressed, by examining the genetic requirements for DSB checkpoint components in this response. Do exogenous, non-meiotic DSBs lead to similar effects, perhaps also in somatic cells, or does this require the context of a delayed synaptonemal complex?

4) Similarly, the manuscript elegantly shows that increased TORC1 activity results in delayed or defective repair of meiotic DSBs, but is lacking in mechanistic detail. Here, examining the localization and activity of repair pathway components could prove insightful. Again, whether these observations reflect a general effect of high TORC1 activity on DSB repair or is limited to germline-specific lesions should be tested, as previous studies have described that low TORC1 activity can impair DSB repair/DNA damage response.

Reviewer #3:

This paper analyzes the interesting intersection between the regulation of metabolism and meiotic progression. It is well known that entry of meiosis in budding yeast is regulated by nutrient availability. The PI has previously published that similar pathways, involving TORC1 and the GATOR1/2 complexes, regulate meiosis in *Drosophila*. This paper is interesting because it connects these regulators of metabolism to double strand break formation as well as p53 and stress responses. The data are convincing and support the conclusions. The most significant results are that DSBs trigger a stress response, which must be modulated by regulating the TORC1 pathway and preventing hyperactivation of p53. The data are interesting with significant impact in areas of regulating gametogenesis and meiotic recombination. There are several issues raised below that are mostly asking for more clarity in describing mutants and cytological results. A couple of experiments are suggested, which entail looking at a couple additional genotypes in the assays presented and thus should not be technically challenging unless there are viability issues. These are meant to generalize the results to the entire pathway presented in Figure 1 but are not essential for the main conclusions of the paper.

In several places are statements that TORC1 activity facilitates entry into meiosis (e.g. subsections “GATOR1 promotes the repair of meiotic DSB” and “GATOR2 opposes a GATOR1 dependent response to meiotic DSBs”). This seems to be mostly taken yeast results. In the absence of GATOR1 in flies, it might be predicted that oocytes either don't form oocytes or meiotic entry does not occur. In fact, Wei et al. reported a delay in meiotic entry in GATOR1 mutants. My question is whether the observed delay in meiotic entry may be more accurately described as a defect in progression. It is also critical to know if the mutants being analyzed in this paper are null alleles. Hypomorphs or RNAi could have leaky phenotypes when it comes to meiotic entry. In subsection “GATOR1 mutants hyperactivate p53 in response to meiotic DSBs”, a *nprl3* is referred to as null, but this is in passing and this information should be much earlier. The authors should be careful to state whether the mutants are null and result in loss of GATOR1 or GATOR2 activity.

Paragraph three of subsection “GATOR1 promotes the repair of meiotic DSB”: Is the persistence of DSBs in region 3 oocytes associated with dorsal-ventral polarity defects? This is observed in DSB repair mutants like *spnA*.

Subsection “GATOR1 mutants hyperactivate p53 in response to meiotic DSBs”: The authors show that GATOR1 mutants hyperactivate p53. To show this is part of the TORC pathway, an experiment with p53 expression in a TORC mutant would be useful. Also, can the authors add p53 be in the model of Figure 6?

Subsection “GATOR1 inhibits retrotransposon expression in*Drosophila*”: The most confusing results concern the relationship between the GATOR1, p53 and TEs. The effect of these mutants on TE expression is DSB dependent. However, while *nprl2/3* mutants have elevated TE expression, they also have elevated, rather than reduced, p53. One explanation is that the TORC pathway is downstream of p53. However, the phenotype of the p53 *nprl3* double is additive with respect to TE expression. As mentioned above, the correct interpretation of these experiments depends on both mutants being null alleles. It also suggests the elevated p53 levels in the *nprl* mutants is not the cause of the increased TE expression. More comment on these discrepancies would be appreciated. It would also be helpful to see a GATOR1-2 (*mio, nprl3*) double mutant, or a *mio* single, to determine if TE regulation is dependent on the pathway.

Subsection “GATOR2 opposes a GATOR1 dependent response to meiotic DSBs”: this seems to be speculation (reasonable, but still a hypothesis) but stated more like fact – that the function of Mio is required to regulate the response to meiotic DSBs. If this were true, the *mei-P22, mio* double would be as fertile as *mei-P22* single. Is that true? Can the authors comment on the small ovary phenotype? When does developmental arrest occur and is that related to some kind of checkpoint? Is it temporally separate from the induction and repair of DSBs?

Subsection “GATOR2 opposes a GATOR1 dependent response to meiotic DSBs”: "…regulation of meiotic DSB repair." and "…meiotic repair of DSBs…"

Subsection “GATOR1 opposes retrotransposon expression”: the implication is that TE expression is upregulated by genotoxic stress, correct? Is that known or a correlation with DSB formation.

Figure 2G gives the impression that many of the mutants have more DSBs in region 2A than wildtype. This could be due to asynchrony and variation within the different pro-oocytes of region 2A. Thus, there should be *nprl* mutant pro-oocytes that have the same number of foci as wild-type, the difference being that as the oocytes progress, the number of foci increases because the DSBs are not repaired. The authors should correct his by showing oocytes of equivalent stage (difficult) or showing all pro-oocytes from one germarium to show how the progression of foci number changes with time.

Figure 2H: Add *spnA* to the graph. There appear to be two differences between the *spnA* and the GATOR phenotypes. First, the γ-H2Av foci persist in the nurse cells as well as the oocyte in *spnA* mutants. Second, 100% of *spnA* oocytes have region 3 foci.

Figure 6 and legend: The figure lacks a panel A. It might be better to put the germarium schematic at the top.

---

## [Author Response]

Reviewer #1:In this manuscript, Wei and colleagues delineate a pathway whereby double-stranded breaks (DSBs) inhibit mTORC1 activity via the GATOR1 complex, and this in turn allows repair of the double-stranded breaks. Overall, this is an interesting finding. A couple of the causality links of this story, however, need to be further tested to make the story solid:1) Figure 1C shows that inhibition of DSB formation leads to increased mTORC1 activity in a mutant condition where mTORC1 activity is aberrantly low due to loss-of-function of GATOR2 (mio^2^ mutation). However, this does not show that in the wildtype condition DSBs affect mTOR activity. If the model is correct, the DSBs that occur normally in the ovary should be inhibiting mTORC1. Removing these DSBs should alleviate this suppression. So does the mei-W68^1^ mutation increase S6K phosphorylation in the ovary compared to wildtype animals?

We performed two sets of experiments to address the reviewer’s question. First, as suggested by reviewers #1 and #2, we examined TORC1 activity in *mei-w68^1^*single mutants by western blot and determined that *mei-w68^1^* mutant ovaries do not have increased TORC1 activity relative to wild-type ovaries (Figure 1). A similar result was observed in *mei-P22* mutants, which also fail to generate meiotic DSBs (Figure 1—figure supplement 3). We believe this observation is not surprising when one considers the anatomy of the *Drosophila* ovary. In wild-type ovaries, meiotic DSBs are present in only a small number of 16-cell cysts in the germarium and are repaired prior to the rapid growth of the egg chamber. Thus, ovarian cysts that contain meiotic DSBs represent an exceedingly small percentage of the tissue in the ovary. Thus, we reasoned it is unlikely that increasing TORC1 activity in only a small number of germarial ovarian cysts would result in an increase in TORC1 activity in the ovary that could be observed by western blot. However, please note that *spnA/Rad51* mutants, which retain meiotic DSBs throughout oogenesis, have reduced TORC1 activity relative to wild-type ovaries as measured by p-S6K levels assayed by western blot. These experiments are described in the second paragraph of the subsection “Mio prevents the constitutive inhibition of TORC1 activity in response to meiotic DSBs”, Figure 1 and Figure 1—figure supplement 2.

To obtain better resolution as to when and where meiotic DSBs impact TORC1 activity during oogenesis, we used an antibody against the phosphorylated form of the downstream TORC1 target 4E-BP. Briefly, we determined that “In contrast to what is observed in wild type, in *mio* mutant ovarian cysts, p4E-BP levels remain low in germline cells in region 2b and beyond (Figure 2C, arrow). Consistent with the western blot analysis, *mio, mei-w68* double mutant ovarian cysts have an approximately three-fold increase in p4E-BP staining in region 2b of the germarium relative to *mio* mutants (Figure 2C, E and H) Notably, the increase in TORC1 activity in the *mio, mei-w68* double mutants is restricted to the germline, consistent with blocking meiotic DSBs having cell autonomous effects on TORC1 activity in the germline. Additionally, consistent with our western blot analysis in Figure 1—figure supplement 2, ovarian cysts from *spnA/Rad51* mutants, which do not repair meiotic DSBs, have reduced levels of p4E-BP staining (Figure 2F). Taken together these data strongly suggest that *mio* is required to oppose/attenuate the downregulation of TORC1 activity triggered by the presence of meiotic DSBs.” These new results are described in the third paragraph of the subsection “Mio prevents the constitutive inhibition of TORC1 activity in response to meiotic DSBs” and Figure 2.

2) The role of GATOR1 in mediating the effect of DSBs on mTORC1 needs to be tested more rigorously using loss-of-function epistasis. From the data in Figure 1C, the authors conclude DSBs affect mTORC1 via GATOR1. However, the data do not exclude the alternate possibility that DSBs and GATOR1 affect mTORC1 via parallel pathways. One way to test this is to assay S6K phosphorylation in ovaries of the following genotypes:- WT- mei-W68^1^- iml1- (GATOR1 loss-of-function)- iml1-, mei-W69^1^ double mutants

As suggested by the reviewer we performed epistasis analysis using the GATOR1 component *nprl3* and *mei-P22* a gene required for the generation of meiotic DSBs. We found that p-S6K levels were not significantly different in wild-type and *mei-p22* mutant ovaries. Moreover, *mei-p22, nprl3* double mutants had lower levels of p-S6K than *nprl3* single mutants. Thus, blocking the formation of meiotic DSBs does not in and of itself result in increased TORC1 activity. This experiment is described in the fourth paragraph of the subsection “Mio prevents the constitutive inhibition of TORC1 activity in response to meiotic DSBs” and summarized in Figure 1—figure supplement 3.

In addition, this manuscript is lacking molecular mechanisms. How does mTORC1 activity affect DSB repair? Is it through delaying cell cycle progression? Is it through affecting transcription or translation of proteins required for repair?

In order to examine why high TORC1 activity delays the repair of meiotic DSBs we examined three downstream targets of TORC1, *atg1*, 4E-BP/*thor* and S6K. Atg1 and 4E-BP are inhibited by TORC1 activity while S6K is activated by TORC1. Our results suggest that the delay in the repair of meiotic DSBs observed in GATOR1 mutants is due, at least in part, to the increased activity of S6K. Our negative results were also informative. We determined that null mutants of *atg1,* which isrequired for autophagy, as well as null mutants of the translational inhibitor 4E-BP/*thor,* do not alter the kinetics of meiotic DSB repair during oogenesis. Thus, we concluded it is highly unlikely that high TORC1 activity delays the repair of meiotic DSBs through the constitutive inhibition of autophagy or the inhibition of the translational inhibitor 4E-BP. These experiments are detailed in the first paragraph of the subsection “Co-depleting S6K rescues the increase in the steady state number of meiotic DSBs in *iml1* germline depletions” and Figure 4.

Alternatively, the upstream mechanism how DSBs affect GATOR1 activity is also missing. This may be a significant amount of work. Hence, it is an editorial decision how much molecular mechanism is wished in the present manuscript.

Although we believe that defining the upstream pathway connecting meiotic DSBs to the TORC1 regulatory machinery is beyond the scope of this manuscript, we did explore the role of the ATM downstream target and checkpoint protein, Chk2. Chk2 is known as Loki in *Drosophila*. Specifically, we determined that removing Chk2/Loki activity partially rescues the *mio* mutant phenotype. From this result we conclude that Chk2/Loki is partially responsible for the downregulation of TORC1 activity observed in *mio* mutants. These results are presented in the last paragraph of the subsection “Mio prevents the constitutive inhibition of TORC1 activity in response to meiotic DSBs” and Figure 1—figure supplement 4.

Reviewer #2:[…] This work is timely and well described, and adds to a growing literature examining the role of metabolism in gametogenesis. Overall the data are lovely and appear well controlled. Additional insight into the mechanisms behind these novel observations would strengthen the manuscript, and experiments that probe these issues in somatic cells would make it of more general interest to eLife readers.1) The authors demonstrate that in mio mutants, disruption of DSB formation leads to increased TORC1 activity, and they repeatedly describe this result as a GATOR-dependent response. However, the observation that TORC1 activity responds to the presence or absence of DSBs in ovaries lacking mio demonstrates that GATOR2 is not required for this response (in contrast to the model depicted in Figure 6), and would suggest that it is likely independent of GATOR1 as well. The role of GATOR1 is not directly tested here, and DSBs could in principle affect TORC1 activity through any one of multiple positive or negative TORC1 regulators/components. Without evidence that GATOR1 is specifically/selectively involved in this signal, I don't see the justification for calling this a GATOR1-dependent effect in the Title, Abstract, etc.

In trying to present all theoretical possible explanations for our data, we presented an overly complicated model that reflected possible roles for different GATOR2 components (such as *wdr24*) that were not examined in the current manuscript. This has now been corrected. The current model shows that Mio is not involved in regulating the initial downregulation of TORC1 activity in response to meiotic DSBs but is required to attenuate or turn off this response. This model is consistent with all of our data. The model is presented in Figure 9 and discussed in the Discussion subsection “The GATOR complex and the response to meiotic DSBs”.

We agree with reviewer #2 that we have not formally proven that the GATOR1/TSC pathway is activated by the presence of meiotic DSBs and have now adjusted the text accordingly. Importantly, we have altered the Abstract and have corrected our language throughout the manuscript to reflect the speculative nature of the connection between meiotic DSBs and the direct activation of a GATOR1/TSC dependent pathway. However, we strongly disagree that the most logical explanation for our results is that the response to meiotic DSBs activates a pathway that is completely independent of the GATOR/TSC pathway. We believe a much simpler explanation is that the regulation of TORC1 activity in the female germline requires two steps: (1) the activation of a GATOR1/TSC dependent pathway by meiotic DSBs and (2) the failure of Mio to attenuate or turnoff this activated pathway. This would be consistent with our published epistasis analysis which demonstrated that double mutants of *mio* and GATOR1 or TSC components rescue the *mio* mutant phenotypes (Wei et al., 2014). It would also be consistent with the observation that GATOR1 is the only identified downstream target of the GATOR2 complex. These ideas are now presented in the Discussion as a model.

I don't see the justification for calling this a GATOR1-dependent effect in the Title.

Please note that we do not reference a GATOR1-dependent effect in the Title.

The title states:

“The GATOR complex regulates an essential response to meiotic double-stranded breaks in *Drosophila*”

We believe the title accurately reflects our data in that (1) the GATOR1 complex impacts the repair of meiotic DSBs and (2) the GATOR2 component Mio is required to attenuate a TORC1 inhibitory pathway that is activated by meiotic DSBs.

2) The requirement for other well characterized TORC1 inputs (e.g., nutrient and insulin pathways) should be addressed.

Our apologies to the reviewer we are not sure what experiment is being suggested here. Please note that TSC, a well-known target of Akt1, is a component of the Insulin signaling pathway. However, we believe the examination of how meiotic DSBs impact insulin/Tor signaling beyond TSC and the GATOR complex are outside the scope of this manuscript.

A more thorough probing of the mechanism by which DSBs affect TORC1 activity should be included here.

Although we believe that defining the upstream pathway connecting meiotic DSBs to the TORC1 regulatory machinery is beyond the scope of this manuscript, we did explore the role of the ATM downstream target and checkpoint protein, Chk2. Chk2 is known as Loki in *Drosophila*. Specifically, we determined that removing Chk2/Loki activity partially rescues the *mio* mutant phenotype. From this result we conclude that Chk2/Loki is partially responsible for the downregulation of TORC1 activity observed in *mio* mutants. These results are presented in the last paragraph of the subsection “Mio prevents the constitutive inhibition of TORC1 activity in response to meiotic DSBs” and Figure 1—figure supplement 4.

The effects of mei-P22 mutation on TORC1 activity should be examined in otherwise wildtype animals and other mutations/conditions that lead to low TORC1 signaling.

We performed two sets of experiments to address the reviewer’s question. First, as suggested by reviewers #1 and #2, we examined TORC1 activity in *mei-w68^1^*single mutants by western blot and determined that, *mei-w68^1^* mutant ovaries do not have increased TORC1 activity relative to wild-type ovaries (Figure 1). A similar result was observed in *mei-P22* mutants, which also fail to generate meiotic DSBs (Figure 1—figure supplement 3). We believe this observation is not surprising when one considers the anatomy of the *Drosophila* ovary. In wild-type ovaries, meiotic DSBs are present in only a small number of 16-cell cysts in the germarium and are repaired prior to the rapid growth of the egg chamber. Thus, ovarian cysts that contain meiotic DSBs represent an exceedingly small percentage of the tissue in the ovary. Thus, we reasoned it is unlikely that increasing TORC1 activity in only a small number of germarial ovarian cysts would result in an increase in TORC1 activity in the ovary that could be observed by western blot. However, please note that *spnA/Rad51* mutants, which retain meiotic DSBs throughout oogenesis, have reduced TORC1 activity relative to wild-type ovaries as measured by p-S6K levels assayed by western blot. These experiments are described in the second paragraph of the subsection “Mio prevents the constitutive inhibition of TORC1 activity in response to meiotic DSBs” and Figure 1 and Figure 1—figure supplement 2.

To obtain better resolution as to when and where meiotic DSBs impact TORC1 activity during oogenesis, we used an antibody against the phosphorylated form of the downstream TORC1 target 4E-BP. Briefly, we determined that “In contrast to what is observed in wild type, in *mio* mutant ovarian cysts, p4E-BP levels remain low in germline cells in region 2b and beyond (Figure 2C, arrow). Consistent with the western blot analysis, *mio, mei-w68* double mutant ovarian cysts have an approximately three-fold increase in p4E-BP staining in region 2b of the germarium relative to *mio* mutants (Figure 2C, E and H) Notably, the increase in TORC1 activity in the *mio, mei-w68* double mutants is restricted to the germline, consistent with blocking meiotic DSBs having cell autonomous effects on TORC1 activity in the germline. Additionally, consistent with our western blot analysis in Figure 1—figure supplement 2, ovarian cysts from *spnA/Rad51* mutants, which do not repair meiotic DSBs, have reduced levels of p4E-BP staining (Figure 2F). Taken together these data strongly suggest that *mio* is required to oppose/attenuate the downregulation of TORC1 activity triggered by the presence of meiotic DSBs.” These new results are described in the third paragraph of the subsection “Mio prevents the constitutive inhibition of TORC1 activity in response to meiotic DSBs” and Figure 2.

We believe that examining how mutations in additional components of the TORC1 machinery are impacted by blocking the formation of meiotic DSBs would be technically challenging, and beyond the scope of this manuscript.

3) The proximal signals linking DSBs to TORC1 downregulation also should be addressed, by examining the genetic requirements for DSB checkpoint components in this response. Do exogenous, non-meiotic DSBs lead to similar effects, perhaps also in somatic cells, or does this require the context of a delayed synaptonemal complex?

Our examination of the role of Chk2 is described above. Additionally, we have previously published that ATR does not link DSBs to TORC1 downregulation in *mio* mutants (Iida and Lilly, 2004).

Whether DSBs in GATOR2 mutants trigger the constitutive downregulation of TORC1 activity in somatic cells will be an interesting area for future analysis.

4) Similarly, the manuscript elegantly shows that increased TORC1 activity results in delayed or defective repair of meiotic DSBs, but is lacking in mechanistic detail.

In order to examine why high TORC1 activity delays the repair of meiotic DSBs we examined three downstream targets of TORC1, *atg1*, 4E-BP/*thor* and S6K. Atg1 and 4E-BP are inhibited by TORC1 activity while S6K is activated by TORC1. Our results suggest that the delay in the repair of meiotic DSBs observed in GATOR1 mutants is due, at least in part, to the increased activity of S6K. Our negative results were also informative. We determined that null mutants of *atg1,* which isrequired for autophagy, as well as null mutants of the translational inhibitor 4E-BP, do not alter the kinetics of meiotic DSB repair during oogenesis. Thus, we concluded it is highly unlikely that high TORC1 activity delays the repair of meiotic DSBs through the constitutive inhibition of autophagy or the inhibition of the translational inhibitor 4E-BP. These experiments are detailed in the first paragraph of the subsection “Co-depleting S6K rescues the increase in the steady state number of meiotic DSBs in *iml1* germline depletions” and Figure 4.

Here, examining the localization and activity of repair pathway components could prove insightful.

We agree that the suggested experiments might provide important information for how TORC1 activity impacts DSB repair. However, because of the lack of available reagents these experiments would be extremely challenging and are beyond the scope of this manuscript.

Again, whether these observations reflect a general effect of high TORC1 activity on DSB repair or is limited to germline-specific lesions should be tested, as previous studies have described that low TORC1 activity can impair DSB repair/DNA damage response.

To assess the general role of the GATOR1 complex in the regulation of DSB repair we examined the repair of γ-irradiation induced DSBs in the somatically derived follicle cells of the ovary by monitoring γ-H2Av staining. From this experiment we determined that high TORC1 activity can impact the repair of DSBs in somatic cell populations. Thus, both inappropriately low and high TORC1 activity can impair DSB repair. These experiments are detailed in the subsection “*nprl3* mutant follicle cells are sensitive to genotoxic stress” and Figure 7.

Reviewer #3:[…] There are several issues raised below that are mostly asking for more clarity in describing mutants and cytological results. A couple of experiments are suggested, which entail looking at a couple additional genotypes in the assays presented and thus should not be technically challenging unless there are viability issues. These are meant to generalize the results to the entire pathway presented in Figure 1 but are not essential for the main conclusions of the paper.In several places are statements that TORC1 activity facilitates entry into meiosis (e.g. subsections “GATOR1 promotes the repair of meiotic DSB” and “GATOR2 opposes a GATOR1 dependent response to meiotic DSBs”). This seems to be mostly taken yeast results. In the absence of GATOR1 in flies, it might be predicted that oocytes either don't form oocytes or meiotic entry does not occur. In fact, Wei et al. reported a delay in meiotic entry in GATOR1 mutants. My question is whether the observed delay in meiotic entry may be more accurately described as a defect in progression.

This is an excellent question. However, at this point in time we do not have the reagents to distinguish between a delay and a defect in progression. I am not certain what assay we could use to distinguish between these two possibilities. Because GATOR1 mutants frequently undergo an extra mitotic division, we hypothesized that GATOR1 mutants delay meiotic entry. Yet, we can’t definitively rule out a defect in progression.

It is also critical to know if the mutants being analyzed in this paper are null alleles. Hypomorphs or RNAi could have leaky phenotypes when it comes to meiotic entry. In subsection “GATOR1 mutants hyperactivate p53 in response to meiotic DSBs”, a nprl3 is referred to as null, but this is in passing and this information should be much earlier. The authors should be careful to state whether the mutants are null and result in loss of GATOR1 or GATOR2 activity.

We thank the reviewer for pointing out this error. The GATOR1 (*nprl2, nprl3* and *iml1*) and *mio* alleles used in this study are all null alleles. We have now made this clear the first time the alleles are discussed.

Paragraph three of subsection “GATOR1 promotes the repair of meiotic DSB”: Is the persistence of DSBs in region 3 oocytes associated with dorsal-ventral polarity defects? This is observed in DSB repair mutants like spnA.

As suggested by the reviewer, we determined that eggs laid by *nprl3* null mutant females do not have DV patterning defects. This may reflect the fact that the repair of meiotic DSBs is delayed but not blocked in *nprl3* mutants. Alternatively, high TORC1 activity may override the translational repression of Gurken that drives the pattern defects observed in DNA repair mutants such as *spnA*. These results are presented in subsection “GATOR1 promotes the repair of meiotic DSB”.

Subsection “GATOR1 mutants hyperactivate p53 in response to meiotic DSBs”: The authors show that GATOR1 mutants hyperactivate p53. To show this is part of the TORC pathway, an experiment with p53 expression in a TORC mutant would be useful.

As noted by reviewer #2, previous work has established that some TORC1 activity is required for DSB repair (Ma et al., 2018). Thus, we would predict that TORC1 mutants would activate p53 in the female germline. Additionally, we have previously demonstrated that TORC1 null mutants result in a very early arrest of ovarian cyst development, often prior to the final mitotic cyst division (Wei et al., 2014). Moreover, we have observed that a small fraction of TORC1 mutants cysts undergo apoptosis. Finally, in unpublished data we see a slight activation of p53 in some GATOR2 mutants, including *mio,* which may be the result of low TORC1 activity. For all of these reasons, we believe the result from the suggested experiment would be difficult to interpret.

Also, can the authors add p53 be in the model of Figure 6?

I think this would be difficult to do without causing confusion. We simplified our previous model at the request of the other reviewers. Currently, we believe that p53 is activated in GATOR1 mutants due to the delay in the repair of meiotic DSBs.

Subsection “GATOR1 inhibits retrotransposon expression in *Drosophila*”: The most confusing results concern the relationship between the GATOR1, p53 and TEs. The effect of these mutants on TE expression is DSB dependent. However, while nprl2/3 mutants have elevated TE expression, they also have elevated, rather than reduced, p53. One explanation is that the TORC pathway is downstream of p53. However, the phenotype of the p53 nprl3 double is additive with respect to TE expression. As mentioned above, the correct interpretation of these experiments depends on both mutants being null alleles. It also suggests the elevated p53 levels in the nprl mutants is not the cause of the increased TE expression. More comment on these discrepancies would be appreciated.

We appreciate the reviewer’s comments and have now clearly stated that the epistasis analysis between *p53* and *nprl3* was performed with null alleles. We realized that the introduction to these experiments was extremely confusing. We have now reorganized as well as rewritten components of this section of the manuscript.

Our data indicate that p53 is necessary but not sufficient for repression of retrotransposon expression in the female germline. This result is consistent with our epistasis analysis, which demonstrated that GATOR1 and p53 function to suppress retrotransposon expression via independent pathways (Figure 8). Because they are functioning in independent pathways, increasing p53 activity would not be predicted to rescue the GATOR1 phenotypes.

As currently outlined in the discussion subsection “GATOR1 opposes retrotransposon expression”, our data support the model that genotoxic stress, due to the delay in the repair of meiotic DSBs, is upstream of both increased p53 activity and derepressed retrotransposon expression.

It would also be helpful to see a GATOR1-2 (mio nprl) double mutant, or a mio single, to determine if TE regulation is dependent on the pathway.

It is unclear what one might predict for a *mio* single mutant, decreased retrotransposon expression perhaps? Because we have not yet examined how *mio* and other GATOR2 components impact genome stability, we believe these experiments are beyond the scope of the manuscript. As noted above, low TORC1 activity, disrupts the response to DSBs in mammalian cells. Consistent with these reports, in unpublished data we see a slight activation of p53 in some GATOR2 mutants, including *mio*. Thus, we believe it would be difficult to interpret the results from these experiments.

Subsection “GATOR2 opposes a GATOR1 dependent response to meiotic DSBs”: this seems to be speculation (reasonable, but still a hypothesis) but stated more like fact – that the function of Mio is required to regulate the response to meiotic DSBs. If this were true, the mei-P22 mio double would be as fertile as mei-P22 single. Is that true?

To address this question, we examined *mio, mei-w68* double mutants. We determined that blocking the formation of meiotic DSBs partially rescues the fertility deficit in *mio.* We propose the partial nature of the rescue may reflect the requirement for *mio* at multiple times during oogenesis as well as the decreased fertility associated with blocking the formation of meiotic DSBs. Notably, the GATOR2 complex has been reported to play a role in spindle assembly and thus may impact the construction of the efficiency of the meiotic divisions (Platani et al., 2015). Please note we tried to be conservative in the interpretation of this result stating: “*…*we demonstrate that the tissue specific requirement for Mio during oogenesis is due, at least in part, to the generation of meiotic DBSs during oogenesis.” These results are presented in subsection “Mio prevents the constitutive inhibition of TORC1 activity in response to meiotic DSBs” and Figure 1—figure supplement 1.

Please note, in *mio* mutantsDSBs trigger a small egg chamber phenotype. These data indicate that *mio* is required to regulate a physiological response to meiotic DSBs. We do not claim that *mio* directlyregulates the repair of meiotic DSBs.

Can the authors comment on the small ovary phenotype. When does developmental arrest occur and is that related to some kind of checkpoint. Is it temporally separate from the induction and repair of DSBs.

We observe problems with oocyte specification in *mio* mutants beginning in late region 2a of the germarium (Iida and Lilly, 2004). However, while the oocyte fails to maintain the oocyte fate, other aspects of egg chamber development continue for some time including the endoreplication of nurse cell nuclei. Thus, it is difficult to establish exactly when *mio* egg chambers arrest their development. We have not been able to temporally separate the induction of meiotic DSBs and the failure to properly maintain oocyte fate in *mio* mutants. Both these events occur as early as region 2a.

We explored the role of the ATM downstream target and checkpoint protein, Chk2. Chk2 is known as Loki in *Drosophila*. Specifically, we determined that removing Chk2/Loki activity partially rescues the *mio* mutant phenotype. From this result we conclude that Chk2/Loki is partially responsible for the downregulation of TORC1 activity observed in *mio* mutants. These results are presented in subsection “Mio prevents the constitutive inhibition of TORC1 activity in response to meiotic DSBs” and Figure 1—figure supplement 4.

Additionally, we have previously published that ATR does not link DSBs to TORC1 downregulation in *mio* mutants (Iida and Lilly, 2004).

Subsection “GATOR1 opposes retrotransposon expression”: the implication is that TE expression is upregulated by genotoxic stress, correct? Is that known or a correlation with DSB formation.

It has been noted by Barbara McClintock and others, that genotoxic stress activates retrotransposons although the exact mechanism of activation remains elusive (Beauregard, Curcio, and Belfort, 2008; Bradshaw and McEntee, 1989; Hagan, Sheffield, and Rudin, 2003; Walbot, 1992). We proposed that “germ line cells may be at an increased risk for transposon derepression due to the genotoxic stress associated with meiotic recombination.” To test the model that unrepaired meiotic DSBs trigger retrotransposon expression we examined retrotransposon transcript levels in *spnA/Rad51* mutant ovaries and determined that they were increased relative to wild type (Figure 8). These data are consistent with work from HeLa cells that LINE1 element activity is increased in Rad51 depleted cells (Liu et al., 2018). This work is presented in subsection “GATOR1 inhibits retrotransposon expression in *Drosophila*” and Figure 8.

Figure 2G gives the impression that many of the mutants have more DSBs in region 2A than wildtype. This could be due to asynchrony and variation within the different pro-oocytes of region 2A. Thus, there should be nprl mutant pro-oocytes that have the same number of foci as wild-type, the difference being that as the oocytes progress, the number of foci increases because the DSBs are not repaired. The authors should correct his by showing oocytes of equivalent stage (difficult) or showing all pro-oocytes from one germarium to show how the progression of foci number changes with time.

We found that GATOR1 mutants had an increase in the number of γ-H2Av foci relative to wild type. This is not just an impression. We used a modified method of counting γ-H2Av foci that was originally presented in Mehrotra and McKim (Mehrotra and McKim, 2006). For both mutant and wild type germaria, we reported the number of foci in the ovarian cyst that had the highest levels of γ-H2Av staining in the oocyte in region 2A of the germarium. As can be seen in Figure 3, this number can be the same as is observed in wild type germaria. However, on average, the number of foci in the GATOR1 mutants is higher. GATOR1 mutant germarium are somewhat misshapen, at least in part, because of the large number of 32 cell cysts. Thus, it would be difficult to establish equivalent stages in the germarium in mutant and wild type. Additionally, it is not clear how scoring all of the oocytes within a cyst, which would be a large amount of work, might alter our interpretation of the role of GATOR1 in the response to meiotic DSBs.

*Figure 2H: Add spnA to the graph. There appear to be two differences between the spnA and the GATOR phenotypes. First, the* γ*-H2Av foci persist in the nurse cells as well as the oocyte in spnA mutants. Second, 100% of spnA oocytes have region 3 foci.*

We did not quantify the retention of γ-H2av foci phenotype, γ-H2av foci also persist in the nurse cells in GATOR1 mutants although this is to a lesser degree then observed in *spnA* mutants (Figure 3). As suggested by the reviewer we added *spnA* mutants to the graph in Figure 3I-K.

Figure 6 and legend: The figure lacks a Panel A. It might be better to put the germarium schematic at the top.

We have altered and simplified the model as suggested by several reviewers (Figure 9).